# A preregistered multi-lab replication of Maier et al. (2014, Exp. 4) testing retroactive avoidance

Markus A. Maier[1]*, Vanessa L. Buechner[1], Moritz C. Dechamps[1], Markus Pflitsch[1], Walter Kurzrock[1], Patrizio Tressoldi[2], Thomas Rabeyron[3,4], Etzel Cardeña[5], David Marcusson-Clavertz[5,6], Tatiana Martsinkovskaja[7]

1 Department of Psychology, Ludwig-Maximilians University, Munich, Germany, 2 Science of Consciousness Research Group, Dipartimento di Psicologia Generale, Università di Padova, Padova, Italy, 3 Department of Psychology, University of Lorraine, Nancy, France, 4 Psychology Department, KPU, University of Edinburgh, Edinburgh, United Kingdom, 5 Department of Psychology, Lund University, Lund, Sweden, 6 Department of Psychology, Linnaeus University, Växjö, Sweden, 7 Institute of Psychology, Russian State University for Humanities, Moscow, Russia

* markus.maier@psy.lmu.de

**Data Availability Statement:** The data underlying the results presented in the study are available from https://osf.io/yqqfz/files/

## Abstract

The term "retroactive avoidance" refers to a special class of effects of future stimulus presentations on past behavioral responses. Specifically, it refers to the anticipatory avoidance of aversive stimuli that were unpredictable through random selection after the response. This phenomenon is supposed to challenge the common view of the arrow of time and the direction of causality. Preliminary evidence of "retroactive avoidance" has been published in mainstream psychological journals and started a heated debate about the robustness and the true existence of this effect. A series of seven experiments published in 2014 in the *Journal of Consciousness Studies* (Maier et al., 2014) tested the influence of randomly drawn future negative picture presentations on avoidance responses based on key presses preceding them. The final study in that series used a sophisticated quantum-based random stimulus selection procedure and implemented the most severe test of retroactive avoidance within this series. Evidence for the effect, though significant, was meager and anecdotal, Bayes factor ($BF_{10}$) = 2. The research presented here represents an attempt to exactly replicate the original effect with a high-power ($N$ = 2004) preregistered multi-lab study. The results indicate that the data favored the null effect (i.e., absence of retroactive avoidance) with a $BF_{01}$ = 4.38. Given the empirical strengths of the study, namely its preregistration, multi-lab approach, high power, and Bayesian analysis used, this failed replication questions the validity and robustness of the original findings. Not reaching a decisive level of Bayesian evidence and not including skeptical researchers may be considered limitations of this study. Exploratory analyses of the change in evidence for the effect across time, performed on a post-hoc basis, revealed several potentially interesting anomalies in the data that might guide future research in this area.

**Funding:** DM was supported by the SPR Research Fund from the Society for Psychical Research. https://www.spr.ac.uk/home The funders had no role in study design, data collection and analysis, decision to publish, or preparation of the manuscript.

**Competing interests:** The authors have declared that no competing interests exist.

## Introduction

In 2014, the *Journal of Consciousness Studies* published an article by Maier and colleagues [1] that reported seven experiments testing the effects of randomly selected stimulus presentations on behavioral decisions that had occurred before those presentations. Specifically, they tested the influence of randomly drawn future negative picture presentations on avoidance responses preceding them. Both picture processing and response selection were maintained at nonconscious levels. The authors hypothesized that if individuals anticipate the—albeit random—future outcome of a binary choice, they will unconsciously select the less aversive outcome. In four of the seven experiments, the authors observed the predicted effect, while two showed null findings and one a statistical trend. In a meta-analytic summary of the seven studies, they reported a significant but small overall effect, ES = 0.07, $z = 3.79$, $p < .0001$; combined Bayes factor ($BF_{10}$) = 293 (extreme evidence for $H_1$), indicating greater-than-chance avoidance of negative picture presentations. The empirical quality of the data was highest in the final experiment of the series, which applied a sophisticated method involving a quantum-based random number generator for response key and picture assignment and included a powerful sample size. In this rigorous study, a significant effect was also observed, but only with a $BF_{10} = 2$, considered as merely anecdotal evidence for $H_1$. Despite the cumulative results, this latter finding cast some doubt on the reproducibility of this effect. Thus, an exact and powerful replication was performed to clarify this and will be reported herein.

The phenomenon addressed in the studies mentioned above is called "retroactive avoidance" and belongs to a broader class of psychological responses denoted by the terms "precognition" or "retroactive influence". Similar effects from a similar design using unconscious picture processing but conscious response selection have been reported by Bem ([2] Exp. 2), and evidence for the existence of precognition was documented in a recent meta-analysis by Bem, Tressoldi, Rabeyron, and Duggan [3] that included 90 experiments run between 2001 and 2013, many of which were direct replications of one of Bem's [2] studies. This meta-analysis suggested that an effect could primarily be located within paradigms that involve nonconscious stimulus processing, so-called "fast thinking" protocols, although no overall effect was observed with slow thinking paradigms, confirming the findings of Galak, LeBoeuf, Nelson, and Simmons [4] and Ritchie, Wiseman, and French [5].

Although Maier et al.'s [1] findings have not been discussed, Bem's original findings have been extensively debated and critiqued. Questionable research practices, such as *p*-hacking and publication bias as a potential source of Type I error, have been addressed by Francis [6] and Schimmack [7]. The usefulness of frequentist methods for analyzing the results, given the low power of the original studies, has been questioned by Wagenmakers, Wetzels, Borsboom, and Van der Maas [8] and Rouder and Morey [9], with both teams suggesting Bayesian analyses (see, however, the re-analysis by Bem, Utts, and Johnson [10], which found partial support using Bayesian analyses). Given these methodological weaknesses, which also affected a large portion of empirical findings in psychology at that time, LeBel and Peters [11] recommended a stronger emphasis on exact replications. Although these criticisms cannot be applied to the work of Maier et al. [1], we agree with this last proposition. In addition to meta-analyses, each effect obtained with one specific paradigm should be tested for replicability within this paradigmatic framework (see also [12]). Since the most conservative study with regard to the randomization procedures used—Maier et al.'s Experiment 4 [1]—revealed an effect that was significant but weak and only anecdotally relevant in terms of Bayesian evidence, we followed this suggestion and pursued a preregistered multi-lab attempt to replicate [1] (Exp. 4) based on Bayesian analyses.

Before we go into a description of the actual study, we will briefly mention the theoretical background of the original work of [1] and of this study in order to provide an explanation for

why the original authors focused on unconscious processing. The original authors derived their hypothesis from a group of interpretations that link the emergence of conscious moments from unconscious processing to quantum theory. Central to their argument were the orchestrated objective reduction (OrchOR) philosophy of mind of Roger Penrose and Stuart Hameroff [13–17] and the generalized quantum theory (GQT) put forward by Harald Atmanspacher, Hartmut Römer, Walter von Lucadou, Thomas Filk, and Harald Walach [18–23]. Both theories link unconscious processing to properties of quantum states and conscious moments to the process of measurement, including the establishment of local reality (note that GQT only applies the mathematical rules of quantum mechanics to these types of processes but makes no assumption about the exact physical nature of these processes). Within the unconscious realm, information processing follows quantum rules that involve superpositions [24], as well as spatial [25] and temporal non-locality [14, 26–29]. The latter could be explained by the hypothesis that the arrow of time may be bidirected at the quantum level and that the directed flow of time, from the past via the present to the future, only occurs in classical reality [14]. These phenomena are nevertheless unrestricted by the Plank constant, and, following [13] Maier et al. [1] argued that–if these interpretations are true- they can be extended to the macroscopic level. Regarding the possibility of empirically observing retroactive avoidance, both theories suppose the existence of such effects when studied with paradigms that test unconscious processing (see also [30–32]). It has to be emphasized that these interpretations of quantum mechanics are favored only by a minority of experts in the field and they are still debated amongst all the other interpretations that have been put forward.

We will now describe the design of the present replication study, which followed the protocol of the original study ([1] Exp. 4) exactly. The main goal of this study was to test the replicability of the retroactive avoidance effect with a high statistical power together with a Bayesian analysis of the results, as suggested by [8]. All procedural details and statistical methods used have been preregistered on OSF and can be found here: https://osf.io/yqqfz. The participating labs all confirmed that they used the protocol exactly as outlined in the preregistration. To make sure that all labs followed the exact same procedure, VLB informed all collaborators through extensive personal (phone or meetings) and email communications about all relevant procedural details. She also sent a written "instruction for experimenters" to all collaborators to make sure that they closely followed the original procedure. The instruction document and the eprime code can be found here: https://osf.io/yqqfz/files/.

All labs used masked experimenters (with one short-term exception; see below) and sent the raw data for analyses to the main author. All incoming data have been subsequently transferred into a sequential Bayesian analysis. The final BF score will be reported as evidence for or against the alternative hypothesis. In addition, post-hoc analyses of the variations in the evidence for the effect across labs and of the overall effect across time will be provided for exploratory reasons.

## Method

All research presented in this article involved human participants, and the protocol was approved by the respective ethical boards of the participating universities. The following ethical boards reviewed and approved the protocol: Ethikkommission der Fakultät 11 (Psychologie & Pädagogik) an der LMU München (Germany); Comitato Etico Della Ricerca Psicologica, Università di Padova, (Italy); Comité d'éthique de l'université de Nantes, Nantes (France); Regionala Etikprövningsnämnden i Lund, (Sweden); Ethical Committee on Psychology, Institute of Psychology named after L.S. Vygotsky, RGGU (RSUH), Moscow, (Russia). Written consent was obtained from all participants. The study was an exact

replication of [1] (Exp. 4), with study details and main analysis preregistered at OSF before the start of data collection. It has to be noted that the text of the preregistration had been created and stored before the beginning of the data collection (in September 2013), however the first author was not aware of the fact that it had to be frozen to complete the procedure. This was done a few months later (in July 2014) without changing any wording of the original text. So basically, the preregistration was finalized after data from 260 participants had already been collected and inspected for the first time. In addition, the study was originally not preregistered explicitly as a multi lab project (but did not exclude this option either).

The experiment was run on a computer to which a response box or keyboard was attached. We tested retroactive avoidance of negative picture presentations. Both the avoidance response and the picture perception were kept unconscious. To reach this goal, in each trial participants were required to press two predefined response keys simultaneously. The response box (or keyboard) was designed so that one of the two keys always triggered first, regardless of the participants' attempts to press the keys at exactly the same time. In this way, the participants unconsciously made a binary choice. For each trial, each key was randomly assigned to either a negative or neutral masked picture presentation (details below) that appeared after the key-press. Our directional hypothesis was that participants unconsciously anticipate the future outcomes of their actions and therefore would avoid negative pictures more often than expected by chance. This would replicate the original findings of [1] (Exp. 4).

## Participants

The participants in this study comprised 2004 undergraduate and graduate students (1,345 Females, 659 Males; mean age = 23.41 years, SD = 6.57) who participated for course credit. We did not specify any exclusion criteria other than basic vision abilities and age of participants being at least 18 years. Of these, 154 participants were attending the Institute of Psychology, RSUH, Moscow (Russia), 235 the University of Padua (Italy), 103 the University of Nantes (France), 99 Lund University (Sweden), and 1,413 Ludwig-Maximilians University of Munich (Germany). They were recruited through the departments' announcement boards, online recruitment platforms, or handouts distributed during class. Students were told that their participation would involve up to three different experiments assessing psychological states, but no further study details were provided in the recruitment information.

## Materials

### Software and computer

The study was conducted using different computers and screens in different labs, all of which were equipped with Windows-run computers. For trial randomization, a quantum-based random number generator (QRNG) from Id Quantique was attached to the computer (see www. idquantique.com). This hardware device has passed both DIEHARD and NIST tests of randomness and is one of the most powerful means of generating true random numbers based on quantum superpositions [33] (see also various certificates from national agencies on their homepage). E-Prime 2.0 or jsPsych 5.0.3 software was used for response registration and picture presentation. Some labs used keyboards and other response boxes (Black Box ToolKit USB response pad, Cedrus RB-740) for response registration. When keyboards were used, the left and right cursor keys served as response registration devices. When response boxes were used, the lower left and right buttons served as response keys.

## Stimuli

The stimulus pictures used were subsets from the International Affective Picture System (IAPS) [34]. Ten extremely negative pictures with a mean valence of 1.73 (SD = 0.27) and ten neutral pictures with a mean valence of 4.90 (SD = 0.27) on a 9-point rating scale obtained from a normative sample were selected (for details, see [1] Appendix).

## Experimenters

Only trained undergraduate research assistants were used as experimenters. They were masked with regard to the study goal and the pictures used in this experiment. For some of the participants from Lund University only, one of the authors (DM) served as experimenter.

## Procedure

Each participant was tested individually or in a group session in a quiet lab room. In group sessions, individuals were separated by small side walls that prevented visual contact between them. Light was dimmed in the rooms. In some sessions, the retroactive avoidance study was the only study performed; in others, it was the final experiment in a series of up to three studies. These pre-studies were standard psychological experiments that varied across labs and time. The experimenters ensured that these were around 15 to 20 minutes in duration. In group sessions, all participants began each experiment at the same time. The focus study began with a written instruction presented on the screen:

> *In the following experiment, you must press two keys on the keyboard (or response box) as simultaneously as possible. You will see this instruction on the monitor's screen:*
>
> *Please press the keys.*
>
> *When you see this instruction, please press both keys as simultaneously as possible!*
>
> *Afterwards, colored stimuli will be presented, which you should simply watch.*

As soon as the participants had read the instructions, the experimenter explained that they should gently place their index fingers on the keyboard's left and right cursor keys or the left and right bottom keys on the response box. The experimenters immediately checked whether the participants had followed this instruction. The response device was placed on the table in front of the participant with the relevant response keys centered in the midpoint of the computer screen. The monitor was placed at a distance of about 50 cm from the participant. Experimenters emphasized that both index fingers should remain lightly touching the response keys throughout the experiment. The experimenters further instructed participants that, once the "Please press keys" command appeared, they should press both keys as simultaneously as possible. Participants were informed that this was not a speed task but that their responses should be spontaneous. After the key-press, they were asked to simply watch the stimuli presented on the screen following the response.

Each trial started with the key-press command presented on the screen. Once a response was performed, the command line disappeared and, after a 430 ms presentation of a black screen, a masked negative or neutral picture was presented. The masked picture presentation consisted of three consecutive stimulus presentations: a masking stimulus presented for 70 ms, followed by the presentation of a negative or neutral picture for 14 ms followed by the same mask for 70 ms. Each negative and neutral picture was combined with an individual mask. The mask was constructed by dividing each original picture into small squares that were

randomly rearranged, forming a stimulus consisting of the same color and lightness properties as the original, but without any content. This effective masking should ensure a subliminal presentation of the original picture and was successfully used in [1]. After the second masking stimulus had disappeared, a 3000 ms black screen inter-trial interval appeared before the next trial was initiated by the key-press command line. A total of 60 trials were presented in this way. The main trials were preceded by three practice trials with neutral images, which helped the participants to familiarize themselves with the task.

Although participants were told to press both keys simultaneously, given the design of the response devices used, one of the two keys always triggered first. Thus, in any given trial either a left or a right key-press was registered, even though participants subjectively performed two-key simultaneous responses. After each response registration, a randomization procedure took place. The QRNG that was connected to each computer randomly created a bit (0/1) during each trial immediately after response registration. Since this QRNG does not operate with a buffer, it was ensured that this actual bit was always created after the key-press. Prior to data collection, an additional randomization was prepared, in which each participant number was linked to a list of 60 bits—one for each trial—using a QRNG. The combination of the pre-stored bit for each trial and the bit actually created after the response registration then defined whether a negative or neutral masked picture appeared afterwards. Negative and neutral pictures were drawn without replacement until all ten pictures from a subset had been drawn (and the process then began again with the same set), leading to a maximum of six presentations of the same picture within a session of 60 trials (6 x 10). In this way, the consequence of each single response could not be classically or algorithmically anticipated by the participant. Any effects potentially observed could then only be explained by retroactive or unconscious precognitive effects from the future. Since QRNG-based picture selection was based on quantum mechanical outcomes, a true source of randomness was used. Null effects should thus lead to 50% negative and 50% neutral picture presentations on average across trials and participants.

## Results

The Results section consists of three parts. In the first, the main analysis tested the evidence for or against the hypothesis that the sample's mean score of negative picture presentations would be lower than chance expectation (50%). This analysis was predefined in the preregistration proposing a one-tailed Bayesian one-sample *t*-test. The second part, there were two additional analyses. First, analyses are reported with wider priors testing the robustness of the effect obtained in the main analysis; second, data from each lab using the original prior are presented separately. These analyses facilitated the exploration of the robustness and variations of the effect across different labs. In the third subsection, three exploratory analyses address the temporal variation of the sequential BF within the original Study 4 of [1] combined with the newly collected data in the study presented here. These analyses explored the temporal change in evidence for the effect across time (from the initial detection of the effect to the later replication attempt), as suggested by [35–37]; see also [23]. These analyses tested non-random fluctuations within the combined data sets against 10,000 simulated data sets. With the exception of the main analysis, no other analyses were part of the preregistration and are therefore purely post hoc and exploratory in nature.

### Main analysis

Since a sequential Bayesian testing approach was applied in this study, the final sample size was not predefined. Instead, an accumulative data collection and analysis strategy using

Bayesian inference techniques for hypotheses testing was used, as suggested by [8]. All 60 trials per participant were handled following the exact same protocol as the original study. That is, a mean score for each participant was computed and subsequently subjected to the Bayesian analysis. This approach allows for data accumulation (i.e., additional respondents can be tested and results added into the dataset) until a specified Bayes factor (BF) for $H_1$ (or $H_0$) has been reached. It also provides the option of ceasing data collection at a predetermined BF. We defined a BF of 10 as the stopping point for evidence for both $H_0$ and $H_1$. This research method is described in [8]; see also [35] for further details). The BF is arguably the best indicator of the evidence for an effect at any moment of data collection. This statistical test can be used sequentially and gives a precise estimation of the probabilities of the two competing hypotheses at each data point. The BF is consistent, which means it will give a more precise answer the more data it considers even if the null hypothesis is true [38]. It describes the relative amount of evidence that the data provide for or against a postulated effect. In this way, the existence ($H_1$) and the non-existence ($H_0$) of an effect can be tested. A BF of 10 or higher is considered to indicate strong evidence for $H_1$ or $H_0$, respectively. For instance, a $BF_{10} = 10$ means that the $H_1$ is ten times more likely to be true than the $H_0$. All participating labs sent their incoming data at least once per semester to the authors from the LMU, who added the new data to the total data set and calculated the actual BF from the overall data collected up to that point. This was repeated over several years, and the sequential change in the BF was observed closely during this time.

To calculate the BF, a probability distribution for effect size must be specified a priori. The effect size of the original study was $d_{cohen} = .1$. Typically, a Cauchy distribution centered around zero with a scale parameter $r$ is used to describe this prior. This distribution ($\delta \sim$ Cauchy $[0, r]$) identifies the likelihood of the data given that there is an effect, $p(\text{data}|H_1)$. Based on the original effect size, an $r$ of 0.1, i.e., $\delta \sim$ Cauchy $(0, 0.1)$, (see also [1]) was chosen a priori and specified in the preregistration. The entire procedure was also approved by Eric-Jan Wagenmakers via personal communication in 2013 (email correspondence from August 19, 2013). A one-sample $t$-test (one-tailed) was performed on a regular basis, at least once at the end of every semester during periods of data collection, to test whether the actual sample's mean score of negative stimuli presentations were below chance (50%). For all Bayesian analyses, the statistical software tools R (Version 3.5.1) and JASP (Version 0.10.1 [39] and previous versions) were used. This was repeated over several years from October 2013 to March 2019. Although at this time the stopping criterion had not been met, all members of the participating research teams agreed to cease data collection primarily because the financial and human resources were exhausted. Consequently, the actual status of evidence for or against the effect will now be reported.

The final Bayesian one-sample $t$-test (one-tailed) with a total of 2004 participants revealed a $BF_{10} = 0.23$ ($BF_{01} = 4.38$) in favor of the null hypothesis. The mean score for negative stimuli for all participants was $M = 29.97$, $SD = 3.92$, providing moderate evidence for a null effect. Thus, against our prediction, the participants' mean score of avoidance responses was not clearly below chance level. Fig 1 represents a sequential analysis of the BF across all participants in the temporal order of testing.

## Additional analyses

To better understand the exact nature of the main result additional analyses were performed that were not part of the original preregistration.

**Robustness analyses.** When testing a directional hypothesis with a Bayesian analysis using a small $r = 0.1$ for the prior, $\delta \sim$ Cauchy $(0, 0.1)$, it is difficult to reach a $BF_{01} > 10$ and thus to confirm the null hypothesis with a reasonable amount of data. Given the postulated

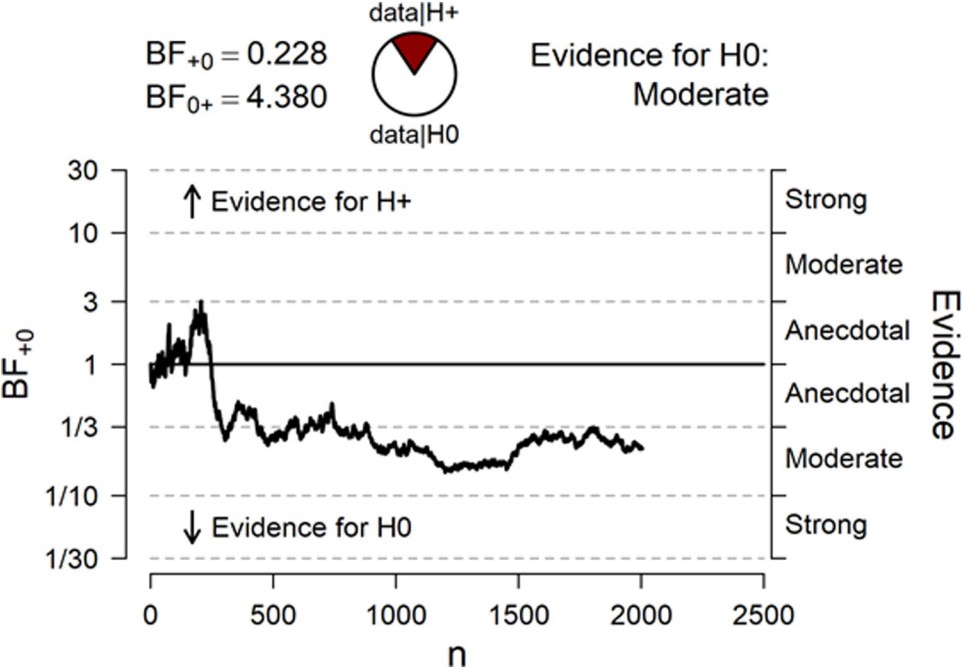

**Fig 1. Sequential Bayes factor curve for all 2004 participants in the temporal order of data collection.**

meager effect size, a much larger sample size than that presented here would be required. Therefore, we also conducted robustness analyses using alternative priors provided by JASP to test the null effect revealed here. The results are illustrated in Fig 2.

As the graph illustrates, when wider priors were applied in the additionally performed Bayesian one-sample $t$-tests, final $BFs_{01} > 30$ were found, indicating very strong evidence for the $H_0$. These post-hoc analyses support our interpretation of the original data analysis that a true null effect was detected there. The raw data can be found here: https://osf.io/yqqfz/files/.

**Variations across labs.** Next, several Bayesian one-sample $t$-tests (one-tailed) with the original prior were performed separately for each lab to test any variations in the effects between the different participating labs. Table 1 presents the sample size, the final BFs, mean scores, and standard deviations for each lab.

As the table above indicates, none of the individual labs produced strong evidence for or against the effect. The strongest trend in line with the prediction was observed at the lab in Moscow University. By contrast, the two most powerful sub-samples (from Padua and Munich) show moderate evidence for $H_0$. To test the homogeneity of the results across labs, we also calculated a meta-analysis with a random-effects model. The heterogeneity estimates were not significant (tau = .0007; $I^2$ = .01%; Q(4) = 4.36, p = .36), indicating a rather homogenous set of data. Mean overall effect size was ES = .008 (SE = .02, p = .76) also revealing a null finding.

In sum, the non-significant heterogeneity test and the low sample size in most sub-samples prevents any further interpretation of the variations. However, given the high sample size in the German data collection one could assume that this partial result might be strongest related to the true population parameter.

## Exploratory analyses

In the following section exploratory analyses are provided on a post-hoc basis to test a theoretical proposition regarding potential temporal variations of the retroactive avoidance effect. This

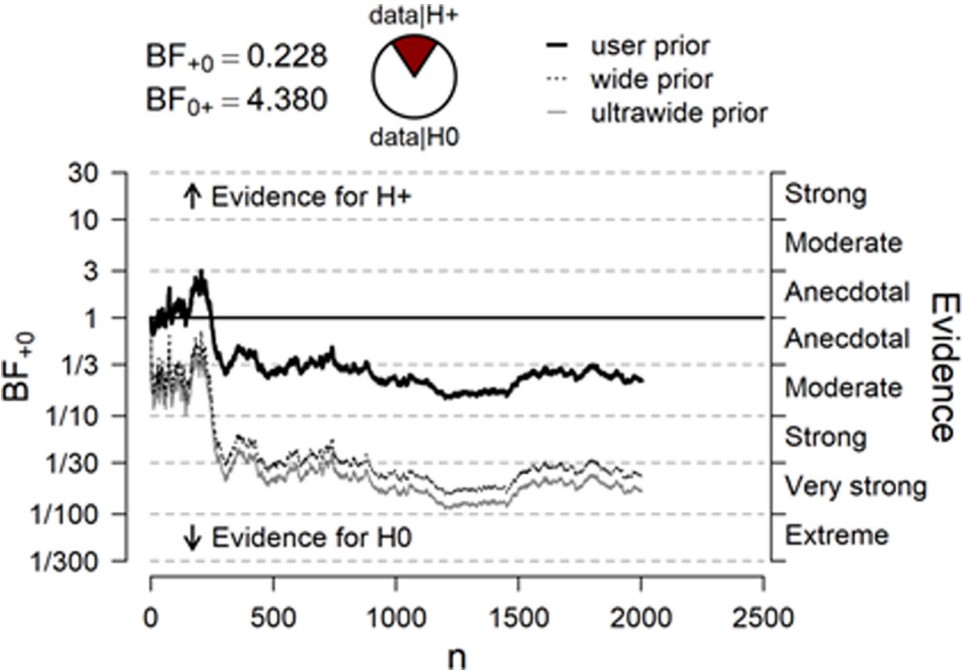

**Fig 2. Sequential Bayes Factor curves for different priors used.**

important theoretical aspect was not considered by Maier et al. [1] nor the preregistration of the study presented here. This issue is central to GQT and attracted our closer attention only in 2018, a year before this study ended. Regarding retroactive avoidance, GQT and its special variant of quantum mechanics' no-signal theorem, the non-transmission axiom (NTA) [23] supposes only the existence of time-symmetric entanglement correlations rather than time-reversed causal signals [20, 23]. Otherwise, the effects of future events on past information processing would violate the Second Law of Thermodynamics [40] and certain restrictions based on special relativity, such as the impossibility of supra-luminal signal transfer [41]. As entanglement correlations behave indeterministically, von Lucadou et al. [23] in their model of pragmatic information (MPI) proposed that, when evidence for retroactive effects has been found in a first study, in later attempts at replication, these effects should disappear as a result of long-term unsystematic variations and lead to a decline or displacement of the overall effect. Moreover, the complementarity relation between effect detection and its future replication prevents the systematic use of retroactive effects at a classical level. Maier, Dechamps, and Pflitsch [35] built on these propositions by arguing that, in spite of these difficulties of replication, a

**Table 1. Bayesian and descriptive analyses of the retroactive avoidance effect for the five individual participating labs.**

|  | $N$ | $BF_{10}$ | Evidence | *Mean* | *SD* |
|---|---|---|---|---|---|
| Germany | 1413 | 0.20 ($BF_{01} = 5$) | moderate for $H_0$ | 30.00 | 3.88 |
| Italy | 235 | 0.31 ($BF_{01} = 3.26$) | moderate for $H_0$ | 30.12 | 4.10 |
| Russia | 154 | 1.93 ($BF_{01} = 0.52$) | anecdotal for $H_1$ | 29.49 | 3.83 |
| France | 103 | 1.19 ($BF_{01} = 0.84$) | anecdotal for $H_1$ | 29.52 | 4.27 |
| Sweden | 99 | 0.40 ($BF_{01} = 2.50$) | anecdotal for $H_0$ | 30.20 | 3.80 |

Sub-samples are ordered according to descending sample size.

systematic change of evidence could be observed for these effects across time. This additional assumption would help distinguish MPI dependent decline effects from simple regressions to the mean effects. They suggested that the temporal change of the effect across time might indeed follow a systematic oscillation pattern, whereas the overall mean score of replications should not statistically deviate from chance expectation (see also [37]). In this way, all assumptions of quantum mechanics, including the randomness postulate, would be fulfilled, and scientific evidence for such effects would be shifted from an analysis of the samples' mean score against chance to systematic temporal oscillations. Dechamps and Maier [37] developed three analytical methods to test these assumed systematic oscillations against random fluctuations.

The three methods were designed to test the oscillations of the effect for data that combined the initial effect detection and replication attempts across time. The time course of the corresponding sequential Bayes factor should be analyzed with: (a) an identification of the highest reach BF found at any time during the data collection compared with the highest BFs reached in 10,000 simulations of the data obtained from the same QRNG used in the original design; (b) a test of the area under the sequential BF (energy of the curve) with BF = 1 as baseline compared to the 10,000 simulations; and (c) fast Fourier transforms (FFTs) of the sequential BF of the human data and the 10,000 simulations with a comparison of the amplitudes obtained. These three analyses test the non-random variation of the effect across time and provide a conservative test of non-random fluctuations within such data sets. Again, we wish to emphasize that this theoretical background and analytical methods were not available during the planning stage of the present study but were only developed during the last year. The following analyses are therefore purely exploratory and are proposed here for testing during future research into effects of this nature.

**Temporal analyses.** We performed three post-hoc analyses to achieve a better understanding of the effects' development over time [37]. For these analyses, we included the initial data and our later replication data. These were the only two studies to have been run with a QRNG within this specific retroactive avoidance paradigm and were the only studies relevant to the theory outlined above, since only quantum-based random mechanisms were addressed in this research framework. To this end, we examined the sequential Bayesian analyses of all retroactive avoidance data obtained from these two studies, i.e. a combination of the 324 data files from [1] (Exp. 4; unfortunately, three original E-Prime files were corrupt and could not be used for the present analysis; the original $n$ was 327) and the 2,004 participants in the replication attempt, which were arranged in the exact temporal order of data collection. We compared this dataset of 2,328 participants to 10,000 simulated datasets of the same size. That is, these simulations consisted of 139,680 random bits each (2,328 participants * 60 trials) aggregated in the same fashion as the experimental data. Subsequently, 10,000 sequential Bayesian $t$-tests with the same parameters as the experimental data (one-tailed; $\delta \sim$ Cauchy $(0, 0.1)$) were conducted based on 2,328 data points each. These simulations represent an experimental null-effect data set. Alpha error probability was set at the .05 level for all analyses.

**Maximum BF.** First, we compared the highest reached Bayes factors obtained from the human sample to those obtained from the 10,000 simulations. The highest BF in the human sample was 48.68 and was reached early on at $n = 65$ (see Fig 3). Only 1.24% of all simulations reached such or a higher BF at any point (see Fig 4A).

**BF energy.** Next, we examined the overall orientation of the BF curve. We calculated the area between the curve and the borderline of evidential power between $H_0$ and $H_1$ at $BF = 1$. A positive value of this area—also called the curve's energy—indicates an overall tendency for the BF to be directionally positioned toward $H_1$. The energy of the human sample's sequential BF was 1453.04, which is surpassed only by 4.2% of the simulations (see Fig 4B). The mean energy of all simulations was found to be $M = -844.42$ ($SD = 8288.72$).

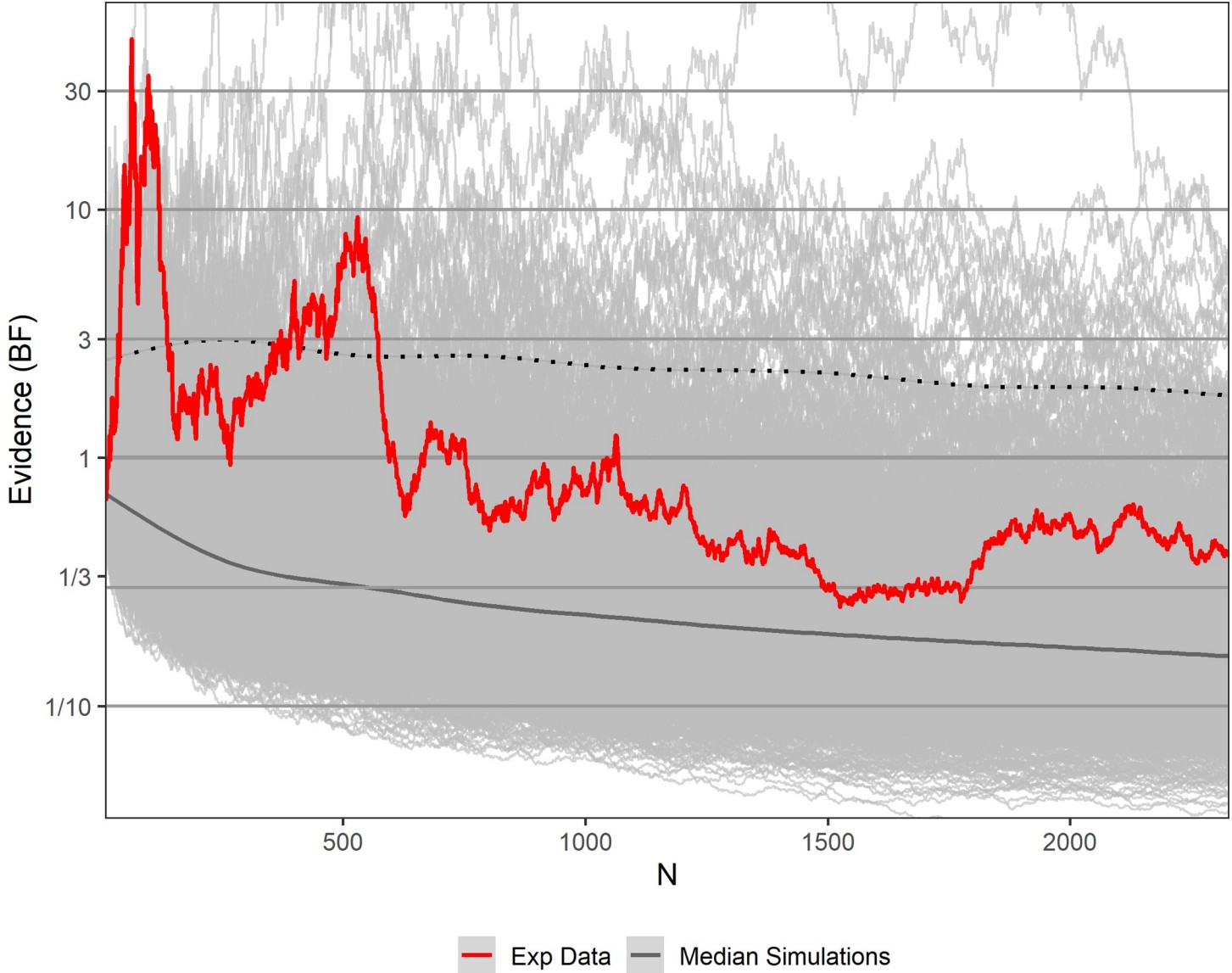

**Fig 3. The sequential BF across 2328 participants (red line) compared to 10,000 sequential BFs obtained from simulated data (gray lines).** The grey line indicates the median of all simulations. 95% of the sequential BFs obtained from all simulations lie below the dotted line.

Admittedly, the results from the maximum BF and the BF energy analyses are to some extent correlated, since extreme BF values are usually accompanied by higher energies and vice versa. To visualize this relation, we added a scatter plot with maximum BF(log-transformed) and BF energy (inverse-hyperbolic-sine-transformed) on the two axes (Fig 5). Each single data point in the graph displays the combination of the respective results of both analyses for all simulations (grey dots) and the human data (red dot). The data cloud indicates how both methods are related to each other. As can be seen, the outcomes of both analyses are positively correlated and extreme scores are very rare. In addition, the graph locates the human data within the cloud of simulated data: within the combined scores the human data set is outstanding ($p = .0104$; that is, only 104 of the simulations reached the same or exceeded its combined score; see blue area in Fig 5). The scatter plot in Fig 5 also indicates that ocassionally

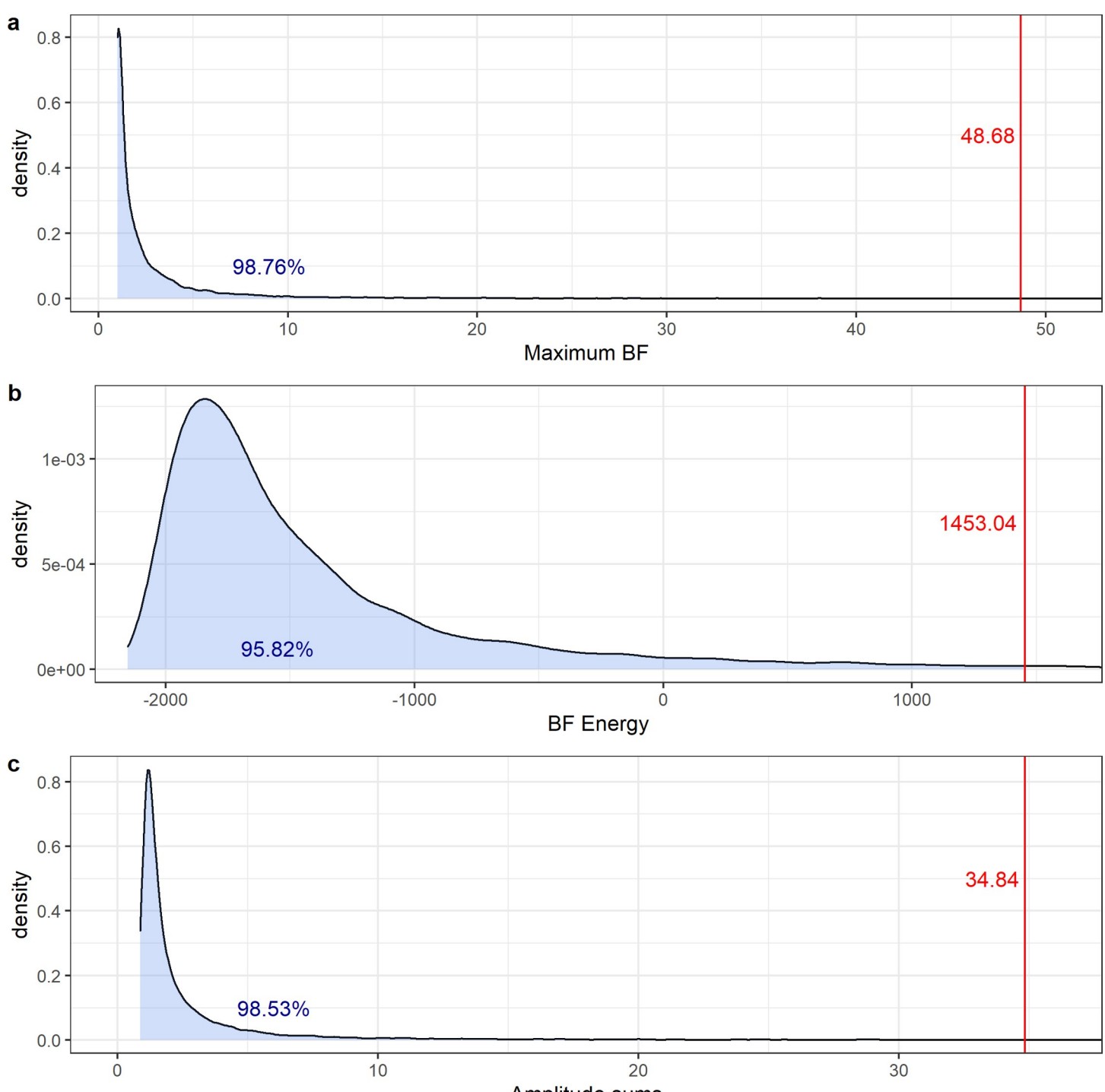

**Fig 4.**  a-c: a) Maximum BF obtained from the human data (located by the red line) and from 10,000 simulations displayed as a density graph representing the maximum BF null distribution (dark line); b) BF energy obtained from the human data (located by the red line) and from 10,000 simulations displayed as a density graph representing the energy BF null distribution (dark line); c) Sum scores of amplitudes obtained from the FFTs of human data (located by the red line) and of 10,000 simulations displayed as a density graph representing the sum score amplitudes null distribution (dark line).

some time series did not exhibit extreme BFs but reached a high level of energy and vice versa. This underscores the usefulfness of reporting both analyses separatly (see above).

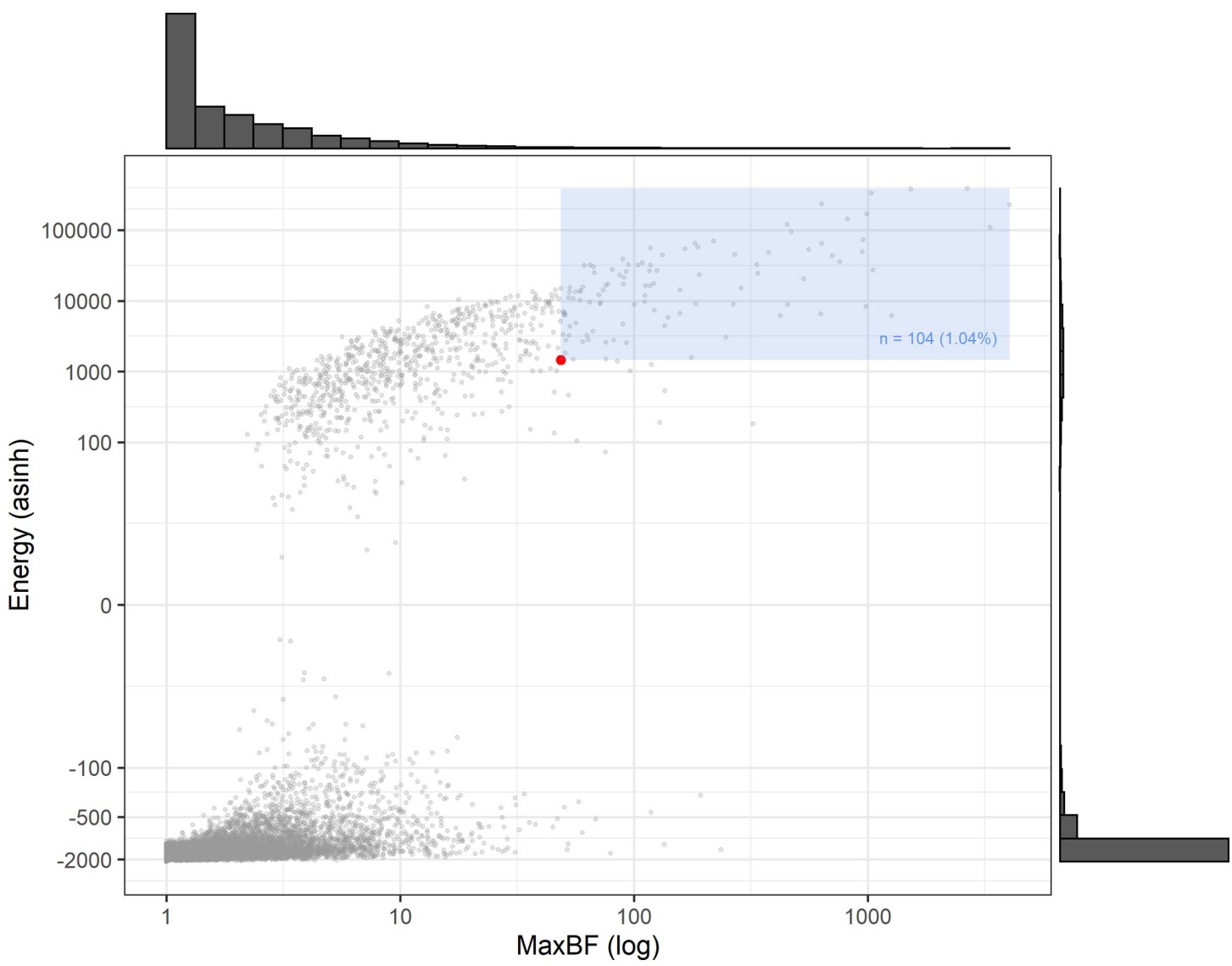

**Fig 5. Combined scores of maximum BF (log-tranformed) and BF energy (inverse-hyperbolic-sine-transformed) based on analyses of human data (red dot) and 10,000 simulations (grey dots).** The blue area indicates the number of combined scores obtained from simulations that reach or exceed the human combined score.

**Frequency spectrum analysis by fast Fourier transform (FFT).** In the third analysis, we decided to examine the oscillation pattern of the sample's sequential BF more closely. Any input signal can be converted to a representation of its composite frequencies via a Fourier transformation. This transform indicates the size of the amplitudes of all frequencies that comprise the input sequence. For a random sequence, none of the frequencies should stand out. Noticeable spikes, however, indicate the presence of a periodic element. An FFT was conducted on the sequential Bayesian analyses of the human data and on each of the 10,000 simulations. Sampling rate was 1/N in each case. Since the resulting transform is symmetric, only the first half is considered in the analysis, resulting in 1,164 tested frequencies. To test the FFT results from the human data against chance occurrence, all 1164 amplitudes obtained from the FFT of the human data set were then added up creating a sum score of the amplitudes obtained from all tested frequencies of this set. In the same way for each of the 10,000 simulations the

sum score of amplitudes was computed (Dechamps and Maier [37] used a similar but less elegant test). The distribution of the sum scores of amplitudes across all simulations served as the null distribution (see Fig 4C). The sum score of amplitudes of the human data set was $Sum_{amp}$ = 34.84. Only 147 of the simulations (1.47%) reached a sum score of 34.84 or higher.

As Fig 4C illustrates, the sum score of amplitudes of the human data (red line) is at the extreme end of the density distribution (dark line). The human sample FFT revealed a more pronounced amplitude pattern than 98.53% of the simulations.

In sum, analyses of the temporal change of the effect across time, expressed by the sequential BF analysis revealed across three different types of methods, show a clear non-random variation of the effect across time. The BF curve variation of human data across time seems to be more pronounced than those obtained from most simulated data sets. This fluctuation in the human data creates an anomaly that fits with von Lucadou et al.'s [23] propositions and is in line with Dechamps and Maier's predictions [37]. The data files including the simulation data and the original data from Maier et al. (Exp. 4) [1], the R code and the jsPsych code can be found here: https://osf.io/yqqfz/files/.

## Discussion

The main goal of this study was to test the replicability of a retroactive avoidance effect observed by Maier et al. [1] (Exp. 4), in a preregistered, high-power, multi-lab study ($N$ = 2,004). With the aid of Bayesian analysis, we tested the hypothesis that randomly presented future negative events can be anticipatorily avoided at an unconscious level. Although the predefined stopping rule of BF > 10, which would indicate strong evidence in favor of either $H_1$ or $H_0$, was not reached, the $BF_{01}$ found within the actual data was 4.38. This provided moderate evidence in favor of the null hypothesis. Given these results, the null hypothesis was four times more likely than the alternative hypothesis to be true. Thus, our prediction was not substantiated, and the results reported by Maier et al. [1] (Exp. 4, ES = 0.1) were not replicated in the present study. Although the moderate evidence leaves sufficient room for speculation that the effect might be confirmed should additional data be collected, the research team in Germany is pessimistic in this regard, though other authors consider that using selected participants, enhancing motivation, and so on may yield different results, as indicated by previous research [42, 43]. First, the actual sequential Bayesian curve displays a rather flat line starting at the second half of the study with a trend toward growing evidence for $H_0$, and it seems unlikely that this trend will turn again in the opposite direction. Second, applying wider priors to the Bayesian analysis revealed strong evidence for the null effect ($BFs_{01}$ > 30). Thus, the fact that we did not reach our actual stopping criterion is due to the prior chosen rather than to a lack of power. We come to the conclusion, therefore, that, based on the overall mean score of avoidance reactions, no retroactive influences were detectable in this experiment.

This result is in line with earlier studies that were unable to replicate retroactive influences at a conscious level using other paradigms (see e.g., [4, 5], partly [3]). It should be noted, however, that these replication attempts failed when they used slow thinking paradigms that involved more conscious procession during response preparation and future event perception, whereas studies using fast thinking protocols have yielded better results [3]. The problem with these latter replication studies, however, is that they lack statistical power. Thus, the present study is, to our knowledge, currently the only study to test precognition using a fast thinking protocol that meets all the criteria for a rigorous, scientifically convincing replication, according to [8], and this replication found evidence for a null effect. However, we encourage researchers to test each paradigm that has been supported in the past with a similar approach as that used here. Powerful replications, as proposed by [12] and [8], would provide additional

evidence to the meta-analyses already available [3] and will offer a clearer picture of the replicability of results in these particular precognition or retroactive avoidance research paradigms. The importance of pre-registered replications was recently supported by an analysis of 15 domains in psychology in which meta-analysis estimates of effect sizes were on average almost three times larger than the pre-registered replications estimates [44].

In addition, the null result observed here is also in line with the assumptions made in standard quantum mechanics and special relativity theory, formulated as no-signal theorem arguing that macroscopic retroactive effects are impossible (for an overview see [45]). Furthermore, theories combining conscious and unconscious processing with computational concepts of quantum theories, such as GQT [18, 20–23], view retroactive avoidance as entanglement correlations that obey the indeterminacy principle in line with the no-signal theorem. Lucadou et al. [23] therefore explicitly argue for an almost zero likelihood of successfully replicating such effects, although these effects should occur unsystematically. However, they propose a special method—the matrix design—that could allow to maintain significant results thanks to the effect's better degree of freedom [46].

Dechamps and Maier [34] (see also [32, 33]), by taking the theoretical constraints of the no-signal-theorem into account, recently proposed an extension of the GQT that supposes systematic non-random oscillations of the evidence for the effect across time. The results are also in line with previous observations regarding replicability in the field of psi research [47]. Bierman [48] describes it as "negative reliability"; Beloff [49] speaks of psi as "actively evasive"; Pallikari and Boller [50] mention a "balancing effect" between positive and negative replications; and Hansen [51] has proposed a broader theory called "the trickster" to explain negative results of this nature. Thus, rather than testing the mean score against chance, future methods might instead focus on FFTs and similar procedures that test systematic oscillation-like variations of evidence for the effect across time (see also [52]). The temporal analyses provided here on a post-hoc basis yielded promising evidence in this direction. An interpretation of the results obtained in the study would be that retroactive avoidance appears, disappears, and even turns into opposite trends from time to time and on different scales (on various frequencies across a broad spectrum; see FFT results above) when additional data are added. This cyclic pattern may be typical for psi effects in general, as indicated by [37, 53]. We wish to emphasize that these analyses do not provide a confirmation of GQT and its extension at this point since they are purely post-hoc in nature. In addition, such a cyclic pattern might also be produced by chance. We quote a reviewer's argument here: "I'll propose that it seems much more plausible that the observed pattern of significant original studies followed by replication failures could be produced if there is zero retroactive avoidance effect in the population. The original significant observations were possibly due to some combination of chance occurrence, flexibility in data analysis, and publication/file drawer bias (note: I think it's quite easy for all of these things to occur even with the best intentions unless you specifically safeguard against them–which is why it's so important that the authors pre-registered the present project)–and the replication failures are simply correctly identifying a true null effect. This explanation has, in general, been demonstrated as quite plausible in simulations (e.g., [12]), and indeed is a major driving force in terms of the current overhaul in methods in social psychology and other fields."We admit that such an argumentation is quite convincing and provides a serious challenge to our interpretation of the temporal variation of the effect. All we can say for now is that the original study, although not preregistered, worked with an a priori planned sample size and, due to the fact that it was a final study conceptually replicating a series of six previous ones, possibilities for biased reporting were somewhat limited. In addition, the FFT results provide a quite unusual oscillation pattern that can hardly be obtained by chance as indicated by a comparision with the FFT patterns found within 10,000 simulations. One might therefore

wonder whether a false positive in a first study and a true negative in a replication might also produce such an outstanding pattern of FFT results. However, to settle that discussion a fully preregistered confirmatory study predicting a temporal change of the effect across time would be needed. Hence, these results should encourage researchers to follow this approach by conducting empirical research that will a priori address time-dependent effect changes in a confirmatory way. Research is always based on trial and error, and exploratory analyses like these can expand our horizons and open up new promising avenues of scientific exploration.

## Acknowledgments

DM was supported by the SPR Research Fund from the Society for Psychical Research.

## Author Contributions

**Conceptualization:** Markus A. Maier, Markus Pflitsch.

**Data curation:** Vanessa L. Buechner, Moritz C. Dechamps.

**Formal analysis:** Markus A. Maier, Vanessa L. Buechner, Moritz C. Dechamps, Walter Kurzrock.

**Funding acquisition:** David Marcusson-Clavertz.

**Investigation:** Vanessa L. Buechner, Moritz C. Dechamps, Walter Kurzrock, Patrizio Tressoldi, Thomas Rabeyron, Etzel Cardeña, David Marcusson-Clavertz, Tatiana Martsinkovskaja.

**Methodology:** Markus A. Maier, Markus Pflitsch.

**Project administration:** Vanessa L. Buechner.

**Software:** Vanessa L. Buechner, Moritz C. Dechamps.

**Supervision:** Markus A. Maier.

**Visualization:** Moritz C. Dechamps.

**Writing – original draft:** Markus A. Maier.

**Writing – review & editing:** Vanessa L. Buechner, Moritz C. Dechamps, Patrizio Tressoldi, Thomas Rabeyron, Etzel Cardeña, David Marcusson-Clavertz, Tatiana Martsinkovskaja.

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
