## [Decision Letter · Decision Letter 0]

16 Apr 2020

PONE-D-20-02005

A Preregistered Multi-Lab Replication of Maier et al. (2014, Exp. 4) Testing Retroactive Avoidance

PLOS ONE

Dear Dr. Maier,

Thank you for submitting your manuscript to PLOS ONE. After careful consideration, we feel that it has merit but does not fully meet PLOS ONE’s publication criteria as it currently stands. Therefore, we invite you to submit a revised version of the manuscript that addresses the points raised during the review process.

Thank you for sending us this manuscript. Sorry for the delay in reviewing it but one of the initial reviewers did not answered despite his commitment to review it. 3 reviewers accepted to review the paper and all did a nice job in reviewing it. I want to thank them here and I'm very grateful for the important insights they provided on your manuscript. I'm sure that you will be able to improve it following all there comments. Please also make sure that you are following all the appropriate reporting guidelines (an adapted checklist may be useful).  

To enhance the reproducibility of your results, we recommend that if applicable you deposit your laboratory protocols in protocols.io, where a protocol can be assigned its own identifier (DOI) such that it can be cited independently in the future. For instructions see: http://journals.plos.org/plosone/s/submission-guidelines#loc-laboratory-protocols

We look forward to receiving your revised manuscript.

Kind regards,

Florian Naudet, M.D., M.P.H., Ph.D.

Academic Editor

PLOS ONE

Journal Requirements:

2. Thank you for including your ethics statement: "All research presented in this article involved human participants, and the protocol was approved by the respective ethical boards of all participating universities.

Written consent was obtained from all participants."

Reviewers' comments:

Reviewer's Responses to Questions

**Comments to the Author**

1. Is the manuscript technically sound, and do the data support the conclusions?

Reviewer #1: Partly

Reviewer #2: Yes

Reviewer #3: Partly

2. Has the statistical analysis been performed appropriately and rigorously? 

Reviewer #1: No

Reviewer #2: Yes

Reviewer #3: Yes

3. Have the authors made all data underlying the findings in their manuscript fully available?

Reviewer #1: No

Reviewer #2: Yes

Reviewer #3: Yes

4. Is the manuscript presented in an intelligible fashion and written in standard English?

Reviewer #1: Yes

Reviewer #2: Yes

Reviewer #3: Yes

5. Review Comments to the Author

Reviewer #1: In this article Maier et colleagues attempt a direct replication of a pre-cognition study published in 2014.

This study tries to establish if the subliminal presentation of a picture with emotionally negative content presented at a given time can reduce participants’ responses to select which picture (negative VS neutral) to view EVEN BEFORE it has been decided which “key press” will be associated with which type of picture.

This work involved 5 different labs and is based on a large sample of participants acquired over several years.

This study was preregistered and the results contain a fairly clear distinction between the confirmatory part (the replication proper) and its exploratory part.

The confirmatory analysis is based on the use Bayes factor to measure of the relative strength of the evidence in favor of the null hypothesis (future presentation of negative stimuli have no effect on the choice of which key will be associated to which stimulus) compared to the alternative hypothesis (future presentation of negative stimuli reduce the chance that the “selected key” will be associated with a negative stimulus – and thus we should lead to lower than chance proportion of aversive stimuli being presented). The prediction being directional the authors use a directional Bayesian one-sample t-test.

Data acquisition was stopped before reaching the pre-registered thresholds because of lack of resources. Based on the available data and some post-hoc analysis the authors conclude that a true null effect was detected.

The exploratory part of the analysis relies on trying to find anomalies in the time series generated by the sequential analysis Bayes factor and by relying on a frequentist approach and using simulations to generate the null distribution.

Part of the data presented is available on OSF.

In general the manuscript is fairly clear, and I generally agree with the results of the confirmatory analysis. The exploratory part presents one major issue (lack of correction for multiple comparison) that should be fixed (see the exploratory section below). I also have some suggestions to make both on the form and the content of the manuscript.

MAIN ISSUES

- lack of correction for multiple comparison in the exploratory analysis

- too much emphasis put on the very speculative theoretical underpinnings of the work

Exploratory analysis

In this part the authors apply what amounts to a frequentist approach to detect surprising results. They rely on simulations to produce the null distribution against which to test their data.

I suggest that the authors mention what is the statistical threshold they use to establish significance? It seems to be 5% but I do not see it mentioned in the manuscript.

I suspect that the 2 first exploratory analysis done by the authors are partly redundant and do not provide independent lines of evidence. Curves with higher Bayes factor will depart further from the BF=1 line and hence have a higher area. I recommend that to better visualize this, authors use a scatter plot of the simulations for “maximum BF VS area” and to display where the data from this study actually lie in this scatter plot.

The main issue in this section lies in the 3rd analysis and to a similar extend.

The 3rd analysis tests 1164 frequencies and in consequence, unless I missed something, shows a need for multiple comparison correction. However I am unclear as to how the authors corrected for this? Without this the risk for false positive might be clearly inflated. Because of the smoothness of the data, 2 neighboring frequencies are unlikely to be independent data points, so I suspect that applying a Bonferroni correction with a significance threshold of 0.000042955 (0.05 / 1164) will be deemed over conservative. If so I suggest that the authors look for inspiration into the statistical methods literature related to EEG to maybe try to find ways on how to best correct their tests.

The first exploratory analysis suffers from a similar problem even though it is not as obvious. By drawing a 95 confidence interval and looking for data points in the BF time series that “stand out”, the authors are implicitly running a statistical test at every time point so more than 2000 statistical tests: this too requires a correction for multiple comparison. Here again the comments made in the previous paragraph regarding data smoothness and finding the right level of correction apply, but they are complicated by the fact that we are dealing with a time series and that autocorrelation might also be an issue.

On a different note, I am wondering why the author did not include an analysis similar to the harmonic oscillation approach used by Dechamps and Maier (2019)?

Comments on theoretical underpinning

In general I would suggest that the authors spend a lot less time on the theoretical speculations and inspirations behind this work.

I think that the authors are putting the cart before the horse, and trying to link their work to quantum mechanics. It is very premature to do so before they have even ascertained the existence of the phenomenon that they are trying to explain. I would strongly suggest that the authors move a lot of the links made to quantum mechanics (introduction line 71 to 118 ; results line 331 to 358) to the discussion section.

I took the liberty to consult a physicist colleague on the parts of the manuscript that were outside of my area of expertise. I am adding below some of her comments that I suggest the authors should take into account.

- l. 83 : Non locality is only ever spatial. There is no reason for it to imply a temporal bi-directionality and this idea does not appear the mentioned references (14, 24, 26).

- l 85 : The cited reference does not correspond to what is mentioned in this assertion. The Emperor's New Mind is a book comparing human mathematical reasoning and turing machines and has not link with temporal bidirectionality.

- l. 347 : The author seem to view entropy as a force that acts in nature to counteract specific effects where as it is in fact a descriptive mathematical quantity. Entropy does not exist as such, but is a sort of indicator invented to summarize several factors that participate to a certain level of disorder of a set of molecules [...]. In a closed system, the level of disorder always increases (and along the entropy that quatifies it), and this is due to the low number of ordered configuration that therefore have a low probability, this is not due to some property of the bodies or the motions involved that would be called entropy and that would condition the possible configuration. The entropy represents the increase in disorder, it does not cause it. Similarly, the increase in disorder is a sign of time passing by [...], but this increase in disorder does not cause the passing of time nor does it condition it.

Penrose himself recognize that his ideas on the nature of consciouness are speculative and his ideas are considered as wrong by several experts in several fields. Moreover Orch-OR is only related to a a potential of quantum mechanics in the emergence of consciousness and does not bear any link to temporal bidirectionality.

The GQT does not have any link with temporal bidirectionality either. And calling it a theory is premature given that it is at this stage a mere hypothesis formulated by a small group physicists, untested and unknown from the physicists community. It is also incomplete and unable to account for many of the the fundamental aspects of quantum mechanics, as admitted by the CGT defenders themselves. It is at this stage very premature to base new research on it.

OTHER ISSUES

Data

Using the data on OSF and JASP, I was partly able to reproduce some of the results of the confirmatory part.

The lack of data dictionary explaining what each column in the data left me wondering why I was not getting the same results as the authors until I realized that number reported in the data was the number of neutral pictures presented to each participants.

Similarly I was not able to easily reproduce all the results from the “Variations across Labs” section. This mostly stems from the way the data was coded, in which there was not a uniform way to know which data point comes from which lab.

Similarly it seems that different labs had different file naming conventions and time stamping of their data files. I would suggest that an extra variable is added to keep track of the overall order in which data samples were acquired, especially given the time series aspect of the data seemed important for the exploratory analysis that the authors suggest.

Similarly the filenames of some samples seemed to have been copies from other files or possibly when moving files between folders ( back2neg_neu_deutsch Version2-524-1 (2).txt ) so I would suggest that add some code to their analysis pipeline to quality check and consolidate the dataset to remove the possibility of the inclusion of any duplicates.

Concerning the exploratory part, I would suggest that the authors also share the data resulting of the 10 000 simulated datasets to allow for a computational reproduction of the authors’ results.

Similarly, and I might be mistaken about, but I think that the ALL the 327 data points from Maier 2014 are actually accessible and not just the 324 as reported in the manuscript. I was able to download them by using the information actually provided in Maier 2014 and get a plot of the sequential analysis similar to the figure 3 of the manuscript. In order to avoid future readers to have to do this bit of data archaeology, I would suggest that the the results from Maier 2014 are either mentioned (with a link) on the OSF project or, better, added on the OSF project related to this paper (if only because long shelf life of things put on OSF compared to that of some institutional or personal repository).

Code

For the exploratory part I would suggest that the authors make available the R code used to analyze the data and generate their figures, not only for transparency, but for other future researchers who might want to further explore those results to evaluate whether they are worth investing their time and effort to replicate.

Similarly and to simplify the work of future replications, I would suggest that the authors share the E-prime and jsPsych code to their experiment.

Pre-registration

I noted some deviations between the pre-registration and the manuscript. I would suggest that the authors cross-check the content and flag any deviations they might spot with the reason behind them. The authors can take some inspiration from the SMART preregistration format: https://osf.io/6vhyt/

From what I can see it seems that the multi-site nature of this experiment was decided after the pre-registration. I suggest the authors give more details on what measures were taken to ensure the same procedure were applied in all laboratories.

The stimuli and mask presentation time are slightly different between the pre-registration and the manuscript. Is this because the refresh rate of the monitor was different?

Also, as a side note and because short presentation time seem important in the current study as the stimuli should be presented subliminally, stimuli presentation timing are notoriously inaccurate on Windows machines and the presentation time reported by the experimental softwares should rarely be taken at face value. I would strongly urge the authors that for future works they externally measure with a photo-diode and report the actual timing accuracy they can get on their set-up.

The robustness checks of the results using different priors and comparing the results across labs were not part of the original pre-registration.

So in general I would suggest that the confirmatory and exploratory parts of the results sections be made even more clear:

1. “Main analysis” should be named “Confirmatory analysis”

Report the pre-registered analysis

1.1. Additional analysis

Report robustness and across labs comparisons

2. “Temporal change across time” should be named “exploratory analysis”

The actual time stamp of the pre-registration is 2014-07-02 07:19 PM, so several months after the experiment was started and after 260 participants were tested. This is not necessarily an issue by itself but it should be mentioned for transparency.

Figures / results consistency

Most of the results in the text are expressed as BF01 but the graphs display BF10. Consistency among the 2 would be good.

Minor comments.

In some places the authors use the word “power” where “sample size” would be more appropriate : eg. line 323.

Reviewer #2: This is a nice and detailed study. The authors undertook a careful exercise to test the replicability of previous findings from a retroactive avoidance effects study, now with a significantly larger sample size, and a well thought out Bayesian sequential testing approach. The authors relied on the Bayes Factor (BF), clearly the best index for evidence detection in the Bayesian world (if the BF can be efficiently computed, at all). The scientific (statistical) premise looks adequate. In the end, they found that I do have some comments, and clarifications.

1. With the purpose of reaching a wider audience, the writing style is pretty verbose. Maybe it can be made more crisp?

2. I inderstand the sequential approach may not start with reaching a fixed sample size. But, what's the effect size we are looking into (not clearly specified)? The writeup, on page 13, directly goes into the prior specification of $\\delta$ to be Cauchy; some argument is necessary for the reader on the context behind that, and citing some literature is not enough.

3. In the "variation study" across labs in various countries, the sample size of Germany is considerably higher than the others. With more data, there is the possibility that it will influence the posterior. I don't see any discussion in this regard while explaining Table 1.

4. Manuscript needs to be checked for inconsistencies in spelling, such as "Psychical" research, should be "Physical".

Reviewer #3: Overall, the manuscript presents rigorously collected data and provides good context for the project, including relevant methodological critiques. The authors engaged with these critiques and took appropriate steps to address them, such that the present project has several methodological advantages (pre-registration, open data, and a large multi-national dataset). The project presents a challenge in that many researchers (myself included) would be skeptical a priori of the retroactive avoidance hypothesis, and admittedly I am no expert in many of the topics being presented (quantum mechanics, etc.). However, I followed the developments surrounding the Bem paper quite closely, and overwhelming data could force a re-examination of those beliefs.

A major selling point of the current project, then, is being pre-registered and presenting open data (although, I think a Registered Report – guaranteeing publication regardless of outcome and thus eliminating publication bias -- might be necessary for such a controversial topic). The pre-registration does a good job laying out the procedure, methods, and decision rules for the analysis. It goes a long way to constrain researcher degrees of freedom, but there remain a few places relating to the analysis and treatment of data that could be further constrained in future efforts. For example, how will the 60 responses per participant be handled? What is the specific statistical test being used? Are there any provisions for missing data or exclusions? Will data from all labs be combined into one and analyzed together, or will meta-analysis or similar be used? These dimensions all introduce possible flexibility in data analysis; that said, for the present project I don't see this as a big concern because the reported analyses follow pretty straightforwardly from what is described in the pre-reg.

The procedures and results are well described in the manuscript, however I think it’s very important to upload the files themselves to the OSF for reproducibility and verification (the eprime files, and syntax for the analysis if available). Because the data collection stopped earlier than planned, the resulting estimates leave perhaps some room for debate. However, from my perspective this is quite compelling evidence for an overall null effect and I’m not sure collecting additional participants would be warranted. I also think variation between sites would be conveyed better through a meta-analysis (P 15), instead of eye-balling the results from different sites. This would give you metrics for heterogeneity (tau, Q, and I^2) indicating the level of heterogeneity between sites. But, this is not too central to the paper and likely would result in the same conclusion (e.g., I suspect you’d get heterogeneity estimates that don’t exceed chance). Overall, I found the analysis and presentation of the confirmatory results quite solid.

Perhaps the area I would critique is the exploratory analyses and the logic underlying the temporal change hypothesis (P 16). To me, the logic behind this argument does not seem very compelling. It seems entropy would have to have “memory” to correct the initial violation of thermodynamics with a second violation of the same law in the opposite direction. This bears a surface level similarity to the gambler’s fallacy, although perhaps I’m not understanding on a deep level. In practice, then, participants in earlier stages of the experiment would show retroactive avoidance, whereas participants in later stages of the experiment (or in replications) would be showing the opposite pattern? I realize this is a hypothesis generated since the project began, but if this is the case you could find robust evidence for this retroactive avoidance effect by instead running a Registered Report using a novel paradigm. Regardless, as the authors note this is quite a challenging hypothesis in terms of the implications for current scientific thinking and methods, and would go against much of our current understanding, so I think there would need to be pretty compelling evidence to validate it. But, I’m also not sure it’s my place to be critiquing these explanations or deeming them plausible or implausible given my lack of expertise in the topics being discussed.

I’ll propose that it seems much more plausible that the observed pattern of significant original studies followed by replication failures could be produced if there is zero retroactive avoidance effect in the population. The original significant observations were possibly due to some combination of chance occurrence, flexibility in data analysis, and publication/file drawer bias (note: I think it’s quite easy for all of these things to occur even with the best intentions unless you specifically safeguard against them – which is why it’s so important that the authors pre-registered the present project) – and the replication failures are simply correctly identifying a true null effect. This explanation has, in general, been demonstrated as quite plausible in simulations (e.g., Simmons et al., 2011), and indeed is a major driving force in terms of the current overhaul in methods in social psychology and other fields. The authors acknowledge this possibility, and pre-registered the present project to safeguard against this.

Then, admittedly I had a pretty high bar for evidence going into the exploratory analyses. The authors appropriately identified these analyses as exploratory, and provided suitable cautions about how readers should interpret them.

Exploratory analyses: (P18-20) With the data being arranged temporally, this high BF is found in the data from the Maier et al., study 4 data, that I understand was not pre-registered? This is, I think, what we would expect given that the earlier study reported a significant result. The problem is that we don't know if this original result may have suffered from some of the classic problems we're concerned about in non-preregistered projects (see above). We're assuming the data from the current replication do not suffer from those issues due to the pre-registration (but, note a Registered Report is the only real way to guarantee no publication bias).

Therefore I'm not sure if this analysis is too convincing for a temporal pattern, so much as a pattern where the pre-registered study found smaller (null) effects? There is some discussion of this issue in the RPP paper (OSC, 2015) and this seems more plausible from my perspective. This concern would apply to both Maximum BF and BF energy analyses, and likely also to the FFT analysis, although I'm not very familiar with that technique. Overall, then, I don’t find these exploratory analyses very convincing as to the presence of a temporal pattern.

Overall, I think these data are valuable and contribute to the discussion around the retroactive avoidance hypothesis. I applaud the authors for engaging with the methodological critiques and taking action (e.g., pre-registration and following advice from some critics) to address them in the present project. I think the confirmatory analyses are quite solid, and my criticisms mostly involve the exploratory hypotheses and analysis which I don’t find very compelling – but they are also clearly labelled as exploratory and secondary. I’ll leave a couple more minor notes at the end.

Other notes:

Note that the pre-reg was first created in September 2013, but it wasn’t actually registered (“frozen”) until July 2, 2014. It looks like data collection started in 2013 and ~260 participants were collected prior to that registration. I think the authors should clarify this point (e.g., why wasn’t it registered before data collection started, and were any data examined before the registration?)

Page 3. Please indicate what effect size this is (cohen's d? pearson correlation?), and the exact sample size for ease of interpretation.

Thanks,

Rick Klein

6. PLOS authors have the option to publish the peer review history of their article (what does this mean?). If published, this will include your full peer review and any attached files.

Reviewer #1: Yes: Remi Gau

Reviewer #2: No

Reviewer #3: Yes: Rick Klein

---

## [Author Response · Author response to Decision Letter 0]

11 May 2020

Dear Dr. Dr. Naudet,

dear Reviewers,

we would like to thank you for your valuable comments on our manuscript entitled „A Preregistered Multi-Lab Replication of Maier et al. (2014, Exp. 4) Testing Retroactive Avoidance” submitted for publication to PLOS ONE. We tried to carefully respond to every comment the editor and the three reviewers made. Below you will find our answers to your concerns and a description of how we adjusted the text. We also added some analyses.

We included the original comments in this letter. Please check our responses to every comment after **.

We are looking forward to your response.

Sincerely,

Markus Maier and co-authors

Journal Requirements:

**OK.

2. Thank you for including your ethics statement: "All research presented in this article involved human participants, and the protocol was approved by the respective ethical boards of all participating universities.

Written consent was obtained from all participants."

**We amended the full names from all boards that approved this study in the „Methods“ section.

**OK, done.

Comments to the Author

5. Review Comments to the Author

Reviewer #1: In this article Maier et colleagues attempt a direct replication of a pre-cognition study published in 2014.

This study tries to establish if the subliminal presentation of a picture with emotionally negative content presented at a given time can reduce participants’ responses to select which picture (negative VS neutral) to view EVEN BEFORE it has been decided which “key press” will be associated with which type of picture.

This work involved 5 different labs and is based on a large sample of participants acquired over several years.

This study was preregistered and the results contain a fairly clear distinction between the confirmatory part (the replication proper) and its exploratory part.

The confirmatory analysis is based on the use Bayes factor to measure of the relative strength of the evidence in favor of the null hypothesis (future presentation of negative stimuli have no effect on the choice of which key will be associated to which stimulus) compared to the alternative hypothesis (future presentation of negative stimuli reduce the chance that the “selected key” will be associated with a negative stimulus – and thus we should lead to lower than chance proportion of aversive stimuli being presented). The prediction being directional the authors use a directional Bayesian one-sample t-test.

Data acquisition was stopped before reaching the pre-registered thresholds because of lack of resources. Based on the available data and some post-hoc analysis the authors conclude that a true null effect was detected.

The exploratory part of the analysis relies on trying to find anomalies in the time series generated by the sequential analysis Bayes factor and by relying on a frequentist approach and using simulations to generate the null distribution.

Part of the data presented is available on OSF.

In general the manuscript is fairly clear, and I generally agree with the results of the confirmatory analysis. The exploratory part presents one major issue (lack of correction for multiple comparison) that should be fixed (see the exploratory section below). I also have some suggestions to make both on the form and the content of the manuscript.

MAIN ISSUES

- lack of correction for multiple comparison in the exploratory analysis

- too much emphasis put on the very speculative theoretical underpinnings of the work

Exploratory analysis

In this part the authors apply what amounts to a frequentist approach to detect surprising results. They rely on simulations to produce the null distribution against which to test their data.

I suggest that the authors mention what is the statistical threshold they use to establish significance? It seems to be 5% but I do not see it mentioned in the manuscript.

**We added a corresponding statement in the result section (line 393).

I suspect that the 2 first exploratory analysis done by the authors are partly redundant and do not provide independent lines of evidence. Curves with higher Bayes factor will depart further from the BF=1 line and hence have a higher area. I recommend that to better visualize this, authors use a scatter plot of the simulations for “maximum BF VS area” and to display where the data from this study actually lie in this scatter plot.

**This is true. These first two analyses are not completely independent as reviewer 1 outlined above. However, ocassionally it occurs that some time series do not exhibit extreme BFs but do reach a high level of energy and vice versa. To cover these, both analyses have been provided. As suggested, we added a scatter plot (see Figure 4) to show how both methods are related to each other and to locate the human data time series within the cloud of simulated time series. Also, in the combined score the human data are outstanding (p = .0104; that is, only 104 of 10,000 simulations reach the same or exceed the combined score).

The main issue in this section lies in the 3rd analysis and to a similar extend.

The 3rd analysis tests 1164 frequencies and in consequence, unless I missed something, shows a need for multiple comparison correction. However I am unclear as to how the authors corrected for this? Without this the risk for false positive might be clearly inflated. Because of the smoothness of the data, 2 neighboring frequencies are unlikely to be independent data points, so I suspect that applying a Bonferroni correction with a significance threshold of 0.000042955 (0.05 / 1164) will be deemed over conservative. If so I suggest that the authors look for inspiration into the statistical methods literature related to EEG to maybe try to find ways on how to best correct their tests.

**In our view, alpha inflation is not a relevant issue in the time series analyses since the human data and each of the simulations is treated in the same fashion and thus contain the same amout of potential inflation. If we compare a potentially inflationally biased human data test set with a distribution of equally treated simulation data test sets, this null distribution also contains the inflation bias within its distribution. If our human data are found at the extreme end of this null distribution it must be significantly different from the simulations and thus from chance by controlling for inflation bias (or keeping it constant).

**Let us explain this for the FFT analysis (but our line of reasoning applies in the same way to the other two analyses as well). The FFT calculates the respective amplitude of serveral frequencies for a given human data set. In a first step, these amplitudes are tested for being extreme by comparing the amplitudes of the respective frequencies with all amplitudes of the same frequencies obtained from the simulations that were also analysed by the FFT. Some of the frequencies in the human data are then declared significant, a certain proportion of those by chance due to the many frequencies that have been tested this way and thus due to the inflation bias. In a second step, 1,000 additional simulations are now individually tested against all the others in the same way, obtaining a number of significant frequencies for each of the 1,000 simulations (also very likely inflated due to the number of tests performed). By combining all 1,000 simulation results, the (inflated) numbers of significant frequencies for all the simulated data sets now constitute the null disctribution for the final crucial test. If the number of significant frequencies found within the human data time series is at the extreme end of this distribution (that also controlls for the inflation bias factor), a significant oscillation pattern was found.

**Although inflation bias is not a problem as outlined above, to circumvent the multiple testing approach in the FFT analysis described, we propose a different and more elegant method for comparing the FFT results from the human data with the FFTs from the 10,000 simulated data sets:

**To test the FFT results from the human data against chance occurrence, all 1,164 amplitudes obtained from the FFT of the human data set were added up creating a sum score of the amplitudes obtained from all tested frequencies of this set. In the same way for each of the 10,000 simulations the sum score of amplitudes was computed. The distribution of the sum scores of amplitudes across all simulations now served as the null distribution (see Figure 5). The sum score of amplitudes of the human data set was Sumamp = 34.84. Only 147 of the simulations (1.47%) reached a sum score of 34.84 or higher. We added this new analysis at the end of the “Exploratory analysis” section and replaced the old one.

The first exploratory analysis suffers from a similar problem even though it is not as obvious. By drawing a 95 confidence interval and looking for data points in the BF time series that “stand out”, the authors are implicitly running a statistical test at every time point so more than 2000 statistical tests: this too requires a correction for multiple comparison. Here again the comments made in the previous paragraph regarding data smoothness and finding the right level of correction apply, but they are complicated by the fact that we are dealing with a time series and that autocorrelation might also be an issue.

**We do not think that this argument is correct. We are not testing every BF obtained in the human data sequential analyses against the corresponding BFs from the simulations. Rather we identify the single highest one and compare it with a distribution of single highest BFs found in the simulation analyses. Thus, no multiple testing is involved here.

**But in case we missed something here, our theoretical argument against the problem of inflation would apply here as well (see our response above).

On a different note, I am wondering why the author did not include an analysis similar to the harmonic oscillation approach used by Dechamps and Maier (2019)?

***The harmonic oscillation approach was critised in a comment by Hartmut Grote (Frontiers in Psychology, Section Cognition, 2019). The curve estimation allows for too many degrees of freedom on the side of the analyst and is therefore not a sufficiently enough objective method to determine the oscillative structure of a time series. Also, the significance testing we used was insufficient.

***We therefore looked for more adequate methods to explore oscillation patterns in time sequences. And apparently, the best method for analyzing discrete data of the kind we collected in our research is the FFT. It was developed for exactly that purpose.

Comments on theoretical underpinning

In general I would suggest that the authors spend a lot less time on the theoretical speculations and inspirations behind this work.

**We agree. We deleted large parts of the theoretical background desciption and also deleted redundancies.

I think that the authors are putting the cart before the horse, and trying to link their work to quantum mechanics. It is very premature to do so before they have even ascertained the existence of the phenomenon that they are trying to explain. I would strongly suggest that the authors move a lot of the links made to quantum mechanics (introduction line 71 to 118 ; results line 331 to 358) to the discussion section.

**See response above. We reduced the theoretical background in the intro to a minimum. We just wanted to make sure that the reader understands why we were focusing on unconscious processing.

**At the beginning of the „Exploratory analyses“ we had to describe the theoretical model that led to the proposition of an oscillating effect. We therefore kept part of the theory in place but deleted any redundancies. Otherwise the reader would not understand why we suddenly performed these post hoc analyses.

I took the liberty to consult a physicist colleague on the parts of the manuscript that were outside of my area of expertise. I am adding below some of her comments that I suggest the authors should take into account.

- l. 83 : Non locality is only ever spatial. There is no reason for it to imply a temporal bi-directionality and this idea does not appear the mentioned references (14, 24, 26).

**Indeed 25 and 26 do not refer to temporal locality but to spatial non-locality, which was mentioned a few words earlier. We put 25 and 26 to that part of the sentence. 14 does refer to temporal non-locality and we added some more references [see 27-29]:

**Tressoldi, P. E., Maier, M. A., Buechner, V. L., & Khrennikov, A. (2015). A macroscopic violation of no-signaling in time inequalities? How to test temporal entanglement with behavioral observables. Frontiers in Psychology, 6, 1061.

**Aharonov, Y., Cohen, E., & Elitzur, A. C. (2015). Can a future choice affect a past measurement’s outcome?. Annals of Physics, 355, 258-268.

**Megidish, E., Halevy, A., Shacham, T., Dvir, T., Dovrat, L., & Eisenberg, H. S. (2013). Entanglement swapping between photons that have never coexisted. Physical Review Letters, 110(21), 210-403.

**The physicist colleague is not completely right. Bell‘s original work indeed adressed spatial non-locality (maybe the colleague is refering to this groundbreaking work) but this was later extended to temporal non-locality (see references above).

- l 85 : The cited reference does not correspond to what is mentioned in this assertion. The Emperor's New Mind is a book comparing human mathematical reasoning and turing machines and has not link with temporal bidirectionality.

**It also addresses quantum mechanics, the measurent problem and the arrow of time and their mutual relationship.

- l. 347 : The author seem to view entropy as a force that acts in nature to counteract specific effects where as it is in fact a descriptive mathematical quantity. Entropy does not exist as such, but is a sort of indicator invented to summarize several factors that participate to a certain level of disorder of a set of molecules [...]. In a closed system, the level of disorder always increases (and along the entropy that quatifies it), and this is due to the low number of ordered configuration that therefore have a low probability, this is not due to some property of the bodies or the motions involved that would be called entropy and that would condition the possible configuration. The entropy represents the increase in disorder, it does not cause it. Similarly, the increase in disorder is a sign of time passing by [...], but this increase in disorder does not cause the passing of time nor does it condition it.

**This is a good point. We treated entropy like a force which is actually wrong. What we rather wanted to say is that an unknown information-related law might underlie the no-signal theorem that leads to the oscillation pattern. We deleted the entropy speculation from the manuscript and revised the corresponding statements accordingly.

**On a side note: Penrose actually does argue that an increase in disorder causes the passing of time.

Penrose himself recognize that his ideas on the nature of consciouness are speculative and his ideas are considered as wrong by several experts in several fields. Moreover Orch-OR is only related to a a potential of quantum mechanics in the emergence of consciousness and does not bear any link to temporal bidirectionality.

**It explicitely does. Most straightforwardly in: 

**Hameroff S. How quantum brain biology can rescue conscious free will. Frontiers in Integrative Neuroscience. 2012; 6(1): 93. https://doi.org/10.3389/fnint.2012.00093

**Hameroff is a co-author of the Orch OR model.

The GQT does not have any link with temporal bidirectionality either. And calling it a theory is premature given that it is at this stage a mere hypothesis formulated by a small group physicists, untested and unknown from the physicists community. It is also incomplete and unable to account for many of the the fundamental aspects of quantum mechanics, as admitted by the CGT defenders themselves. It is at this stage very premature to base new research on it.

***Römer and von Lucadou both stated in personal communications that the GQT does provide links to temporal bidirectionality. The key construct in GQT is non-local entanglement correlations that also include temporal non-locality.

OTHER ISSUES

Data

Using the data on OSF and JASP, I was partly able to reproduce some of the results of the confirmatory part.

The lack of data dictionary explaining what each column in the data left me wondering why I was not getting the same results as the authors until I realized that number reported in the data was the number of neutral pictures presented to each participants.

Similarly I was not able to easily reproduce all the results from the “Variations across Labs” section. This mostly stems from the way the data was coded, in which there was not a uniform way to know which data point comes from which lab.

Similarly it seems that different labs had different file naming conventions and time stamping of their data files. I would suggest that an extra variable is added to keep track of the overall order in which data samples were acquired, especially given the time series aspect of the data seemed important for the exploratory analysis that the authors suggest.

***Good point. A time stamp variable has been added.

Similarly the filenames of some samples seemed to have been copies from other files or possibly when moving files between folders ( back2neg_neu_deutsch Version2-524-1 (2).txt ) so I would suggest that add some code to their analysis pipeline to quality check and consolidate the dataset to remove the possibility of the inclusion of any duplicates.

***We checked all files. They were no dublicates. In case the same subject code was mistakenly used twice for different subjects, eprime just added a „(2)“ to the eprime result file name in order not to overwrite the original result file.

Concerning the exploratory part, I would suggest that the authors also share the data resulting of the 10 000 simulated datasets to allow for a computational reproduction of the authors’ results.

**OK, we added the 10,000 simulated data sets to the OSF.

Similarly, and I might be mistaken about, but I think that the ALL the 327 data points from Maier 2014 are actually accessible and not just the 324 as reported in the manuscript. I was able to download them by using the information actually provided in Maier 2014 and get a plot of the sequential analysis similar to the figure 3 of the manuscript. In order to avoid future readers to have to do this bit of data archaeology, I would suggest that the the results from Maier 2014 are either mentioned (with a link) on the OSF project or, better, added on the OSF project related to this paper (if only because long shelf life of things put on OSF compared to that of some institutional or personal repository).

**OK, this can be done. We added the raw data to the OSF.

**We need to emphasize that the data from these three participants cannot be used for the temporal analyses described in our mansucript here. The problem with the three corrupt eprime_result files is that we do not have time stamps for these three participants. Although we do have their mean scores -as the reviewer correctly states- we cannot place them at the correct (!) temporal position on which their data have been assessed (the Maier et al. 2014 subject numbers do not reflect the exact temporal order of data collection). Since our main argument concering the temporal analyses is based on the exact temporal ordering of the participants‘ data during data collection, we decided to omit these three data points from the temporal analyses. Given the sample size, we do not think that including these data (even if it were possible) would make a remarkable difference.

Code

For the exploratory part I would suggest that the authors make available the R code used to analyze the data and generate their figures, not only for transparency, but for other future researchers who might want to further explore those results to evaluate whether they are worth investing their time and effort to replicate.

Similarly and to simplify the work of future replications, I would suggest that the authors share the E-prime and jsPsych code to their experiment.

**OK , done.

Pre-registration

I noted some deviations between the pre-registration and the manuscript. I would suggest that the authors cross-check the content and flag any deviations they might spot with the reason behind them. The authors can take some inspiration from the SMART preregistration format: https://osf.io/6vhyt/

From what I can see it seems that the multi-site nature of this experiment was decided after the pre-registration. I suggest the authors give more details on what measures were taken to ensure the same procedure were applied in all laboratories.

**Indeed that prereg does not specify the study as a multi-site experiment (but it does also not exclude this possibility). We made a corresponding statement in the text (line 125).

 **During the course of data collection we realized that it would have taken too much time to run this study purely on our own. We therefore invited colleagues who might be interested in this project to help us with data collection in their labs.

**Through extensive personal (phone or meetings) and email communication we made sure that the procedure was exactly the same as the one performed in our lab. We also sent a written instruction for experiementers to our collaborators to make sure that they closely followed our instructions. The instruction sheet is now uploaded at OSF.

**We made a few corresponding statements at the end of the introduction section that describe our efforts to ensure that the same procedure was applied in all participating labs.

 **We think extending data collection to a multi-lab project rather stengthens the scientifc value of the study rather than weakening it. We do not consider this shift in data collection strategy a deviation from the original preregistration.

**Despite slight adjustments of the presentations times chosen (see below), we are not aware of any other deviations from the specifications made in the preregistration.

The stimuli and mask presentation time are slightly different between the pre-registration and the manuscript. Is this because the refresh rate of the monitor was different?

**Yes, this was the reason. We are convinced that these adjustments are not responsible for the null effect (and nobody would argue this way).

Also, as a side note and because short presentation time seem important in the current study as the stimuli should be presented subliminally, stimuli presentation timing are notoriously inaccurate on Windows machines and the presentation time reported by the experimental softwares should rarely be taken at face value. I would strongly urge the authors that for future works they externally measure with a photo-diode and report the actual timing accuracy they can get on their set-up.

**Yes, this is a very good point and should also be adressed before any preregistrations in future projects.

The robustness checks of the results using different priors and comparing the results across labs were not part of the original pre-registration.

**Yes, and we clearly labeled them as additional (i.e., non confirmatory) analyses. We consider them to add some important information about how to interpret the main finding.

So in general I would suggest that the confirmatory and exploratory parts of the results sections be made even more clear:

1. “Main analysis” should be named “Confirmatory analysis”

Report the pre-registered analysis

1.1. Additional analysis

Report robustness and across labs comparisons

2. “Temporal change across time” should be named “exploratory analysis”

**We followed this advice and used the proposed headlines in the revised manuscript.

The actual time stamp of the pre-registration is 2014-07-02 07:19 PM, so several months after the experiment was started and after 260 participants were tested. This is not necessarily an issue by itself but it should be mentioned for transparency.

**Done. We added the following statement:

“It has to be noted that the text of the preregistration has been created and stored before the beginning of the data collection (in September 2013), however the first author was not aware of the fact that it also had to be frozen to complete the procedure. This was done a few months later (in July 2014) without changing anything of the original text. So basically the preregistration was fully finalized after data from 260 participants had already been collected and were inspected for the first time. In addition, the study originally was not explicitely preregistered as a multi lab project (but did not exclude this option either).”

Figures / results consistency

Most of the results in the text are expressed as BF01 but the graphs display BF10. Consistency among the 2 would be good.

**We checked for consistency and adjusted the text when needed.

Minor comments.

In some places the authors use the word “power” where “sample size” would be more appropriate : eg. line 318.

**We changed the wording as suggested.

Reviewer #2: This is a nice and detailed study. The authors undertook a careful exercise to test the replicability of previous findings from a retroactive avoidance effects study, now with a significantly larger sample size, and a well thought out Bayesian sequential testing approach. The authors relied on the Bayes Factor (BF), clearly the best index for evidence detection in the Bayesian world (if the BF can be efficiently computed, at all). The scientific (statistical) premise looks adequate. In the end, they found that I do have some comments, and clarifications.

1. With the purpose of reaching a wider audience, the writing style is pretty verbose. Maybe it can be made more crisp?

**We agree and deleted large portions of the text in the introduction. We also checked for redundancies and deleted them (see our reponse to a similar request from reviewer 1).

2. I inderstand the sequential approach may not start with reaching a fixed sample size. But, what's the effect size we are looking into (not clearly specified)? The writeup, on page 13, directly goes into the prior specification of $\\delta$ to be Cauchy; some argument is necessary for the reader on the context behind that, and citing some literature is not enough.

** The effect size of the original study, that we intended to replicate, was dcohen = .1. The specification of the Cauchy distribution used as prior in the replication attempt was based on this initial effect size. We mention this now in the revision (lines 272-276).

3. In the "variation study" across labs in various countries, the sample size of Germany is considerably higher than the others. With more data, there is the possibility that it will influence the posterior. I don't see any discussion in this regard while explaining Table 1.

**We added a corresponding statement at the end of the “Additional analysis“ section.

4. Manuscript needs to be checked for inconsistencies in spelling, such as "Psychical" research, should be "Physical".

**Could not find the term psychical in the text.

Reviewer #3: Overall, the manuscript presents rigorously collected data and provides good context for the project, including relevant methodological critiques. The authors engaged with these critiques and took appropriate steps to address them, such that the present project has several methodological advantages (pre-registration, open data, and a large multi-national dataset). The project presents a challenge in that many researchers (myself included) would be skeptical a priori of the retroactive avoidance hypothesis, and admittedly I am no expert in many of the topics being presented (quantum mechanics, etc.). However, I followed the developments surrounding the Bem paper quite closely, and overwhelming data could force a re-examination of those beliefs.

A major selling point of the current project, then, is being pre-registered and presenting open data (although, I think a Registered Report – guaranteeing publication regardless of outcome and thus eliminating publication bias -- might be necessary for such a controversial topic). The pre-registration does a good job laying out the procedure, methods, and decision rules for the analysis. It goes a long way to constrain researcher degrees of freedom, but there remain a few places relating to the analysis and treatment of data that could be further constrained in future efforts. For example, how will the 60 responses per participant be handled? What is the specific statistical test being used? Are there any provisions for missing data or exclusions? Will data from all labs be combined into one and analyzed together, or will meta-analysis or similar be used? These dimensions all introduce possible flexibility in data analysis; that said, for the present project I don't see this as a big concern because the reported analyses follow pretty straightforwardly from what is described in the pre-reg.

**How the 60 trials per participant were handled followed the exact protocol of the original study. That is, a mean score for each participant was computed and subjected to the Bayesian one sample t-test analysis in a sequential way. We added a corresponding statement at the beginning of the “Main analysis“ section.

**We did not specify any exclusion criteria other than basic vision abilities and age of participants being at least 18 years. We added a corresponding statement in the “Participants“ section.

**From the Baysian analysis procedure it followed that any newly collected data would be transfered into the sequential BF analysis in the temporal order in which the data were collected. No meta-analytic methods were planned.

The procedures and results are well described in the manuscript, however I think it’s very important to upload the files themselves to the OSF for reproducibility and verification (the eprime files, and syntax for the analysis if available). Because the data collection stopped earlier than planned, the resulting estimates leave perhaps some room for debate. However, from my perspective this is quite compelling evidence for an overall null effect and I’m not sure collecting additional participants would be warranted. I also think variation between sites would be conveyed better through a meta-analysis (P 15), instead of eye-balling the results from different sites. This would give you metrics for heterogeneity (tau, Q, and I^2) indicating the level of heterogeneity between sites. But, this is not too central to the paper and likely would result in the same conclusion (e.g., I suspect you’d get heterogeneity estimates that don’t exceed chance). Overall, I found the analysis and presentation of the confirmatory results quite solid.

**Good point. We added the files and documents to the OSF.

**We also calculated a meta-analysis (random-effects model) across labs. Indeed the heterogeneity estimates did not exceed chance. We added a corresponding paragraph in the “Variations across Labs“ section (a fixed-effects model produced the same result).

Perhaps the area I would critique is the exploratory analyses and the logic underlying the temporal change hypothesis (P 16). To me, the logic behind this argument does not seem very compelling. It seems entropy would have to have “memory” to correct the initial violation of thermodynamics with a second violation of the same law in the opposite direction. This bears a surface level similarity to the gambler’s fallacy, although perhaps I’m not understanding on a deep level. In practice, then, participants in earlier stages of the experiment would show retroactive avoidance, whereas participants in later stages of the experiment (or in replications) would be showing the opposite pattern? I realize this is a hypothesis generated since the project began, but if this is the case you could find robust evidence for this retroactive avoidance effect by instead running a Registered Report using a novel paradigm. Regardless, as the authors note this is quite a challenging hypothesis in terms of the implications for current scientific thinking and methods, and would go against much of our current understanding, so I think there would need to be pretty compelling evidence to validate it. But, I’m also not sure it’s my place to be critiquing these explanations or deeming them plausible or implausible given my lack of expertise in the topics being discussed.

I’ll propose that it seems much more plausible that the observed pattern of significant original studies followed by replication failures could be produced if there is zero retroactive avoidance effect in the population. The original significant observations were possibly due to some combination of chance occurrence, flexibility in data analysis, and publication/file drawer bias (note: I think it’s quite easy for all of these things to occur even with the best intentions unless you specifically safeguard against them – which is why it’s so important that the authors pre-registered the present project) – and the replication failures are simply correctly identifying a true null effect. This explanation has, in general, been demonstrated as quite plausible in simulations (e.g., Simmons et al., 2011), and indeed is a major driving force in terms of the current overhaul in methods in social psychology and other fields. The authors acknowledge this possibility, and pre-registered the present project to safeguard against this.

**We agree, this is a highly relevant counterargument/alternative interpretation of this finding. We added this statement into the discussion session (which can be credited to reviewer 3 if he so desires).

Then, admittedly I had a pretty high bar for evidence going into the exploratory analyses. The authors appropriately identified these analyses as exploratory, and provided suitable cautions about how readers should interpret them.

Exploratory analyses: (P18-20) With the data being arranged temporally, this high BF is found in the data from the Maier et al., study 4 data, that I understand was not pre-registered? This is, I think, what we would expect given that the earlier study reported a significant result. The problem is that we don't know if this original result may have suffered from some of the classic problems we're concerned about in non-preregistered projects (see above). We're assuming the data from the current replication do not suffer from those issues due to the pre-registration (but, note a Registered Report is the only real way to guarantee no publication bias).

Therefore I'm not sure if this analysis is too convincing for a temporal pattern, so much as a pattern where the pre-registered study found smaller (null) effects? There is some discussion of this issue in the RPP paper (OSC, 2015) and this seems more plausible from my perspective. This concern would apply to both Maximum BF and BF energy analyses, and likely also to the FFT analysis, although I'm not very familiar with that technique. Overall, then, I don’t find these exploratory analyses very convincing as to the presence of a temporal pattern.

**Good point, see our response above.

Overall, I think these data are valuable and contribute to the discussion around the retroactive avoidance hypothesis. I applaud the authors for engaging with the methodological critiques and taking action (e.g., pre-registration and following advice from some critics) to address them in the present project. I think the confirmatory analyses are quite solid, and my criticisms mostly involve the exploratory hypotheses and analysis which I don’t find very compelling – but they are also clearly labelled as exploratory and secondary. I’ll leave a couple more minor notes at the end.

Other notes:

Note that the pre-reg was first created in September 2013, but it wasn’t actually registered (“frozen”) until July 2, 2014. It looks like data collection started in 2013 and ~260 participants were collected prior to that registration. I think the authors should clarify this point (e.g., why wasn’t it registered before data collection started, and were any data examined before the registration?)

**We explain this inconsistency in the text:

“It has to be noted that the text of the preregistration had been created and stored before the beginning of the data collection (in September 2013), however the first author was not aware of the fact that it also had to be frozen to complete the procedure. This was done a few months later (in July 2014) without changing anything of the original text. So, basically, the preregistration was finalized after data from 260 participants have already been collected and were inspected for the first time. In addition, the study originally was not explicitely preregistered as a multi lab project (but did not exclude this option either).”

Page 3. Please indicate what effect size this is (cohen's d? pearson correlation?), and the exact sample size for ease of interpretation.

**It was cohen’s d = .1. Based on the 327 subjects of the original study. We mention this now in the “Results“ section.

---

## [Decision Letter · Decision Letter 1]

15 Jun 2020

PONE-D-20-02005R1

A Preregistered Multi-Lab Replication of Maier et al. (2014, Exp. 4) Testing Retroactive Avoidance

PLOS ONE

Dear Dr. Maier,

Thank you for submitting your manuscript to PLOS ONE. After careful consideration, we feel that it has merit but does not fully meet PLOS ONE’s publication criteria as it currently stands. Therefore, we invite you to submit a revised version of the manuscript that addresses the points raised during the review process.

Thank you for your good job in revising the paper. Many thanks to the 4 reviewers who assessed this new draft. As you will see, there are minor changes to be made as suggested by "reviewer 1". Please take these comments into account. 

We look forward to receiving your revised manuscript.

Kind regards,

Florian Naudet, M.D., M.P.H., Ph.D.

Academic Editor

PLOS ONE

Reviewers' comments:

Reviewer's Responses to Questions

**Comments to the Author**

1. If the authors have adequately addressed your comments raised in a previous round of review and you feel that this manuscript is now acceptable for publication, you may indicate that here to bypass the “Comments to the Author” section, enter your conflict of interest statement in the “Confidential to Editor” section, and submit your "Accept" recommendation.

Reviewer #1: (No Response)

Reviewer #2: All comments have been addressed

Reviewer #3: All comments have been addressed

2. Is the manuscript technically sound, and do the data support the conclusions?

Reviewer #1: Yes

Reviewer #2: (No Response)

Reviewer #3: (No Response)

3. Has the statistical analysis been performed appropriately and rigorously? 

Reviewer #1: Yes

Reviewer #2: (No Response)

Reviewer #3: (No Response)

4. Have the authors made all data underlying the findings in their manuscript fully available?

Reviewer #1: Yes

Reviewer #2: (No Response)

Reviewer #3: (No Response)

5. Is the manuscript presented in an intelligible fashion and written in standard English?

Reviewer #1: Yes

Reviewer #2: (No Response)

Reviewer #3: (No Response)

6. Review Comments to the Author

Reviewer #1: First I would like to apologize for the time it took me to do this second round of review: this should have been much faster.

I would like to thank the authors for their clarifications, all the reworking and additions to the manuscript. It makes it clearer where the line between results and interpretation lie.

Updating the code and the data (from this and the previous study) on OSF have also clear added values.

My two main suggestions are:

display for figure 3 the null distribution on which the statistical inference is performed

add a readme to the OSF project and ensure that the code runs as intended when downloaded from source

---

**Although inflation bias is not a problem as outlined above, to circumvent the multiple

testing approach in the FFT analysis described, we propose a different and more

elegant method for comparing the FFT results from the human data with the FFTs from

the 10,000 simulated data sets:

**To test the FFT results from the human data against chance occurrence, all 1,164

amplitudes obtained from the FFT of the human data set were added up creating a

sum score of the amplitudes obtained from all tested frequencies of this set. In the

same way for each of the 10,000 simulations the sum score of amplitudes was

computed. The distribution of the sum scores of amplitudes across all simulations now

served as the null distribution (see Figure 5). The sum score of amplitudes of the

human data set was Sumamp = 34.84. Only 147 of the simulations (1.47%) reached a

sum score of 34.84 or higher. We added this new analysis at the end of the

“Exploratory analysis” section and replaced the old one.

**We are not testing every BF obtained in the human data sequential analyses against the corresponding BFs from the simulations. Rather we identify the single highest one and compare it with a distribution of single highest BFs found in the simulation analyses. Thus, no multiple testing is involved here.

**But in case we missed something here, our theoretical argument against the problem

of inflation would apply here as well (see our response above).

---

I thank the authors for explaining this more thoroughly. This helped me understand why I got confused and thought there was a multiple comparison problem. The text did indeed mention what null distribution was used to make the statistical inference. However the figure of the previous version of the article showed a 95% confidence interval envelope that a) was unrelated to the inference made in the text, b) would actually suffer from multiple comparison problem if it were used as a way to threshold and detect extreme data.

The new approach suggested by the author is an improvement as it is more direct and the link between the figure and the text is much clearer.

In the same spirit I would suggest that :

the authors make the text and the figure 3 more congruent and add a figure (or maybe an inset to the figure 3) similar figure 5 showing where the maximum BF values of the human data lies when compared to the distribution of maximums obtained from the simulation.

Also maybe I am still missing something but if only the maximum BF value is taken for the inference how can **other** maximum BF be found in time series as suggested by this sentence:

“Higher BFs were also found with participants numbered 500 to 600 (i.e., 400 in the early stages of the replication study).”

---

The stimuli and mask presentation time are slightly different between the pre-registration and the manuscript. Is this because the refresh rate of the monitor was different?

**Yes, this was the reason. We are convinced that these adjustments are not

responsible for the null effect (and nobody would argue this way).

---

Indeed I was not arguing that way. I was more trying to understand where some of the differences were coming from.

Code

But I admit that the lack of a README did not help in figuring out:

what packages are needed

where unzipped files should go

in which order the scripts should be run as there seem to be some dependencies between them.

I suggest that such basic documentation should be provided.

I was unable to make some of the R code run (which might be partly due to my lack of R knowledge) and in ‘exploratoryanalyses.R’

Error in rbind(deparse.level, ...) :

numbers of columns of arguments do not match

Could the authors make sure that everything runs as intended from simply downloading the OSF project and trying to run it from there? This is usually even better when performed by a colleague “naive” to the study to ensure a bare minimum of portability to make sure the one’s code does not just run on one’s machine.

On that note, I would like to point to the authors (and the editors) this recent initiative that makes sure that submitted code runs as intended: https://codecheck.org.uk/

I am appending below additional comments and suggestions of my physicist colleague.

Quantum theory

l.74 : « group of theories » should be replaced by « group of interpretations » (which don’t have the level of certainty of a theory)

GQT is an interpretation, even if it has the arrogance to call itself a theory, without any experiments to support its hypotheses. It doesn’t have the consensual character of a theory, it is only supported by a few physicists and only based on interpretation from the quantum model.

OrchOR is a philosophy of mind, also based on interpretation and extrapolation from the quantum model to the mind, without any supporting experiments or direct observations. Thus, at l.76, “theory” should be replaced by “philosophy of mind”.

In my opinion, this is not the kind of foundations that one should build assumptions in other fields upon, so I would suggest that you don’t mention quantum mechanics at all. But if you really want to mention it, it should be clear in your formulation that « if these interpretations are true », then there are similarities with your subject of experiment that would indicate that there could possibly be a causal link between the two hypothetical phenomena. In addition, it should also be very clear that these interpretations are (markedly) minor ones and still debated amongst all the others that have been put forward about the quantum model.

Otherwise, there is a high risk that your article will be automatically considered by any physicist who would read it as deliberately deforming the current consensus in physics in order to match its own conclusions, which would be regrettable if physics and psychology are as interdependent as your article would suggest, because in this case this research field could only benefit from collaboration with physicists.

I understand better now what you mean by « temporal non-locality » and I was aware of the kind of phenomenon that it stands for in your article (which appears in delayed-choice experiments). The formulation is a little confusing, because « locality » etymologically refers to a « location » (in space) so maybe it would be clearer to call it « non-temporality » (this is not an official denomination). I’ve found two or three articles that use the same denomination as you do but a few others use « temporal Bell inequalities » or « temporal entanglement », actually it doesn’t have an official name because it’s only an aspect of the intrication phenomenon in general.

The real experimental effect is called « delayed-choice experiments », but the choice to consider that there is a new physical phenomenon (« temporal non-locality » or else) underlying this effect depends on the interpretation. Notably in the standard interpretation it’s just an effect of observation, and in GQT it will be the same because of the role of consciousness in the observation (basically put : in delayed-choice experiments, there are unobservable events which happen before the observation that are correlated to the choice made by the observer while doing the observation later. But in GQT, only the consciousness is capable of producing a measurement, so nothing unobservable ever happens, so there is no correlation, everything is only determined when the observer arrives). This is the only interpretation of quantum mechanics that you refer to (since OrchOR is not really an interpretation of quantum mechanics but rather an application of it to the philosophy of mind). Be careful then because the effect you refer to with « temporal non-locality » doesn’t exist in the interpretation that you refer to. I would suggest that you choose to mention only one of these ideas (non-temporality or GQT) for consistency’s sake. If you’re looking for an interpretation that would consider the delayed-choice experiments as a manifestation of time bidirectionality, there is the transactional interpretation, but this one doesn’t make any reference to consciousness so it cannot be linked to the rest of your work.

I would consider the reference to non-temporality (or “temporal non-locality”) as the weakest part because if this effect is real it would typically be destroyed very quickly by decoherence (the statistical process that makes quantum effects very unprobable and unstable at large scales and usual temperatures), so it would be very unprobable that it could be maintained through a whole room full of air in order to carry retrocausal information all the way from the stimuli to the subject’s brain.

l.84 : « implies » should be replaced by « could be explained by the hypothesis that », because it’s not a necessary implication. As I mentioned, the delayed choice experiments can be interpreted in several ways (I'm sorry for the lack of clarity of this field of physics, we're working on it). It’s also not clear what affirmation the sources 14,27,28,29 are supposed to support at l.84 because some of them support the existence of the physical phenomenon and some of them support the link between quantum mechanics and psychology.

Reviewer #2: (No Response)

Reviewer #3: (No Response)

7. PLOS authors have the option to publish the peer review history of their article (what does this mean?). If published, this will include your full peer review and any attached files.

Reviewer #1: Yes: Remi Gau & Alice Van Helden

Reviewer #2: No

Reviewer #3: No

---

## [Author Response · Author response to Decision Letter 1]

24 Jun 2020

Dear Dr. Dr. Naudet,

dear Reviewers,

we would like to thank you for your additional comments on our revised manuscript entitled „A Preregistered Multi-Lab Replication of Maier et al. (2014, Exp. 4) Testing Retroactive Avoidance” submitted for publication to PLOS ONE. We tried to carefully respond to every comment you made. In this document you will find our answers to your comments and a description of how we adjusted the text (see **). We also revised some of the figures as you suggested.

Thank you for the open and fair scientific discussion we had and your profound work on this paper.

First I would like to apologize for the time it took me to do this second round of review: this should have been much faster. 

**It is fine, thank you for taking this job seriously.

I would like to thank the authors for their clarifications, all the reworking and additions to the manuscript. It makes it clearer where the line between results and interpretation lie. 

Updating the code and the data (from this and the previous study) on OSF have also clear added values. 

My two main suggestions are:

- display for figure 3 the null distribution on which the statistical inference is performed

- add a readme to the OSF project and ensure that the code runs as intended when downloaded from source

**Ok, good point. We added a “readme” file and added a figure (Figure 4 a-c) that shows the null distributions of all three exploratory analyses and where the human data lie.

**Although inflation bias is not a problem as outlined above, to circumvent the multiple

testing approach in the FFT analysis described, we propose a different and more

elegant method for comparing the FFT results from the human data with the FFTs from

the 10,000 simulated data sets:

**To test the FFT results from the human data against chance occurrence, all 1,164

amplitudes obtained from the FFT of the human data set were added up creating a

sum score of the amplitudes obtained from all tested frequencies of this set. In the

same way for each of the 10,000 simulations the sum score of amplitudes was

computed. The distribution of the sum scores of amplitudes across all simulations now

served as the null distribution (see Figure 5). The sum score of amplitudes of the

human data set was Sumamp = 34.84. Only 147 of the simulations (1.47%) reached a

sum score of 34.84 or higher. We added this new analysis at the end of the

“Exploratory analysis” section and replaced the old one.

**We are not testing every BF obtained in the human data sequential analyses against the corresponding BFs from the simulations. Rather we identify the single highest one and compare it with a distribution of single highest BFs found in the simulation analyses. Thus, no multiple testing is involved here.

**But in case we missed something here, our theoretical argument against the problem

of inflation would apply here as well (see our response above).

I thank the authors for explaining this more thoroughly. This helped me understand why I got confused and thought there was a multiple comparison problem. The text did indeed mention what null distribution was used to make the statistical inference. However the figure of the previous version of the article showed a 95% confidence interval envelope that a) was unrelated to the inference made in the text, b) would actually suffer from multiple comparison problem if it were used as a way to threshold and detect extreme data.

**Good point, we deleted the confidence interval (see Figure 3).

The new approach suggested by the author is an improvement as it is more direct and the link between the figure and the text is much clearer.

In the same spirit I would suggest that:

the authors make the text and the figure 3 more congruent and add a figure (or maybe an inset to the figure 3) similar figure 5 showing where the maximum BF values of the human data lies when compared to the distribution of maximums obtained from the simulation.

**Done, see our response above and Figure 4a-c.

Also maybe I am still missing something but if only the maximum BF value is taken for the inference how can **other** maximum BF be found in time series as suggested by this sentence:

“Higher BFs were also found with participants numbered 500 to 600 (i.e., 400 in the early stages of the replication study).”

**Good point, the statement does not match the analysis and was therefore deleted from the text.

The stimuli and mask presentation time are slightly different between the pre-

registration and the manuscript. Is this because the refresh rate of the monitor was

different?

**Yes, this was the reason. We are convinced that these adjustments are not

responsible for the null effect (and nobody would argue this way).

Indeed I was not arguing that way. I was more trying to understand where some of the differences were coming from.

Code

But I admit that the lack of a README did not help in figuring out:

- what packages are needed

- where unzipped files should go

- in which order the scripts should be run as there seem to be some dependencies between them.

I suggest that such basic documentation should be provided.

**Done, see the “readme” document.

I was unable to make some of the R code run (which might be partly due to my lack of R knowledge) and in ‘exploratoryanalyses.R’

Error in rbind(deparse.level, ...) :

 numbers of columns of arguments do not match

Could the authors make sure that everything runs as intended from simply downloading the OSF project and trying to run it from there? This is usually even better when performed by a colleague “naive” to the study to ensure a bare minimum of portability to make sure the one’s code does not just run on one’s machine. 

**OK, we made sure that now everything works. We also made the independent test as suggested and it worked.

On that note, I would like to point to the authors (and the editors) this recent initiative that makes sure that submitted code runs as intended: https://codecheck.org.uk/

**Thank you. This is helpful!

I am appending below additional comments and suggestions of my physicist colleague.

Quantum theory

l.74 : « group of theories » should be replaced by « group of interpretations » (which don’t have the level of certainty of a theory)

GQT is an interpretation, even if it has the arrogance to call itself a theory, without any experiments to support its hypotheses. It doesn’t have the consensual character of a theory, it is only supported by a few physicists and only based on interpretation from the quantum model.

**We agree, this is an important point you make. However, a theory does not need any empirical evidence to be called a theory and consensus is not a feature of a theory either. But we understand what the reviewer wants to say here and we changed the corresponding statement as suggested.

OrchOR is a philosophy of mind, also based on interpretation and extrapolation from the quantum model to the mind, without any supporting experiments or direct observations. Thus, at l.76, “theory” should be replaced by “philosophy of mind”.

**OK.

In my opinion, this is not the kind of foundations that one should build assumptions in other fields upon, so I would suggest that you don’t mention quantum mechanics at all. But if you really want to mention it, it should be clear in your formulation that « if these interpretations are true », then there are similarities with your subject of experiment that would indicate that there could possibly be a causal link between the two hypothetical phenomena. In addition, it should also be very clear that these interpretations are (markedly) minor ones and still debated amongst all the others that have been put forward about the quantum model.

Otherwise, there is a high risk that your article will be automatically considered by any physicist who would read it as deliberately deforming the current consensus in physics in order to match its own conclusions, which would be regrettable if physics and psychology are as interdependent as your article would suggest, because in this case this research field could only benefit from collaboration with physicists.

** We hear you on this and added several statements in the text that put the interpretations into the context of all the other interpretations of QM.

**Just for the record: If someone really wants to put forward an innovative idea at some point one has to challenge the current consensus or parts of it, otherwise no progress will ever be made. Whether our approach was valid or not was put to an empirical test and we were found to be wrong. So, the current consensus remains intact. We make this very clear in our paper especially in the discussion section.

I understand better now what you mean by « temporal non-locality » and I was aware of the kind of phenomenon that it stands for in your article (which appears in delayed-choice experiments). The formulation is a little confusing, because « locality » etymologically refers to a « location » (in space) so maybe it would be clearer to call it « non-temporality » (this is not an official denomination). 

**According to Einstein’s general relativity theory space and time are inseparable. So, locality as the term is used in general relativity (see also Einstein, Podolsky & Rosen, 1935) always includes both time and space.

I’ve found two or three articles that use the same denomination as you do but a few others use « temporal Bell inequalities » or « temporal entanglement », actually it doesn’t have an official name because it’s only an aspect of the intrication phenomenon in general.

The real experimental effect is called « delayed-choice experiments », but the choice to consider that there is a new physical phenomenon (« temporal non-locality » or else) underlying this effect depends on the interpretation. Notably in the standard interpretation it’s just an effect of observation, and in GQT it will be the same because of the role of consciousness in the observation (basically put : in delayed-choice experiments, there are unobservable events which happen before the observation that are correlated to the choice made by the observer while doing the observation later. But in GQT, only the consciousness is capable of producing a measurement, so nothing unobservable ever happens, so there is no correlation, everything is only determined when the observer arrives). 

**Yes, but consciousness can determine later what has happened before and not only what occurs during the observation.

This is the only interpretation of quantum mechanics that you refer to (since OrchOR is not really an interpretation of quantum mechanics but rather an application of it to the philosophy of mind). Be careful then because the effect you refer to with « temporal non-locality » doesn’t exist in the interpretation that you refer to. 

**It does exist, both in GQT and OrchOr as confirmed from the authors of GQT in personal communication and with regard to orchOr also explicitly in the paper of Hameroff (2012):

[26] Hameroff S. How quantum brain biology can rescue conscious free will. Frontiers in Integrative Neuroscience. 2012; 6(1): 93. https://doi.org/10.3389/fnint.2012.00093

I would suggest that you choose to mention only one of these ideas (non-temporality or GQT) for consistency’s sake.

**See our response above. In addition, we are only referring to what Maier et al.’s arguments were in 2014. Thus, we would like to keep the line of arguments as it is. We already shortened it to a minimum of information.

 If you’re looking for an interpretation that would consider the delayed-choice experiments as a manifestation of time bidirectionality, there is the transactional interpretation, but this one doesn’t make any reference to consciousness so it cannot be linked to the rest of your work.

**Thank you for the hint. And yes, we know about TI and there are also attempts to modify it to include consciousness, but we did not want to make the theoretical part bigger (and more speculative) than absolutely needed.

I would consider the reference to non-temporality (or “temporal non-locality”) as the weakest part because if this effect is real it would typically be destroyed very quickly by decoherence (the statistical process that makes quantum effects very unprobable and unstable at large scales and usual temperatures), so it would be very unprobable that it could be maintained through a whole room full of air in order to carry retrocausal information all the way from the stimuli to the subject’s brain.

**Yes, unlikely but not impossible, that is the point. Our kind of work tried to figure out those boundary conditions under which temporal non-locality could be observed macroscopically. Nowadays, superposition states of relatively large objects are found that -given decoherence- would have been considered highly unlikely years ago.

l.84 : « implies » should be replaced by « could be explained by the hypothesis that », because it’s not a necessary implication. As I mentioned, the delayed choice experiments can be interpreted in several ways (I'm sorry for the lack of clarity of this field of physics, we're working on it). It’s also not clear what affirmation the sources 14,27,28,29 are supposed to support at l.84 because some of them support the existence of the physical phenomenon and some of them support the link between quantum mechanics and psychology.

**OK, we made the suggested change. The references address both temporal non-locality in purely physical and psycho-physical contexts and thus

---

## [Decision Letter · Decision Letter 2]

7 Aug 2020

PONE-D-20-02005R2

A Preregistered Multi-Lab Replication of Maier et al. (2014, Exp. 4) Testing Retroactive Avoidance

PLOS ONE

Dear Dr. Maier,

Thank you for submitting your manuscript to PLOS ONE. After careful consideration, we feel that it has merit but does not fully meet PLOS ONE’s publication criteria as it currently stands. Therefore, we invite you to submit a revised version of the manuscript that addresses the points raised during the review process.

**I would like to thank the reviewers for their help in assessing this important manuscript**. Both suggested to accept it in its current form and I'm please to say that I agree, providing minor changes are made in the abstract.

I propose to use a structured abstract (i.e. more focused on the study methods and results).

Importantly, I ask you to delete any discussion about post-hoc analyses in the abstract.

I propose to add a few words in the abstract to highlight the main strengths and also limitations of the paper.

We look forward to receiving your revised manuscript.

Kind regards,

Florian Naudet, M.D., M.P.H., Ph.D.

Academic Editor

PLOS ONE

Reviewers' comments:

Reviewer's Responses to Questions

**Comments to the Author**

1. If the authors have adequately addressed your comments raised in a previous round of review and you feel that this manuscript is now acceptable for publication, you may indicate that here to bypass the “Comments to the Author” section, enter your conflict of interest statement in the “Confidential to Editor” section, and submit your "Accept" recommendation.

Reviewer #1: All comments have been addressed

Reviewer #2: All comments have been addressed

2. Is the manuscript technically sound, and do the data support the conclusions?

Reviewer #1: Yes

Reviewer #2: (No Response)

3. Has the statistical analysis been performed appropriately and rigorously? 

Reviewer #1: Yes

Reviewer #2: (No Response)

4. Have the authors made all data underlying the findings in their manuscript fully available?

Reviewer #1: Yes

Reviewer #2: (No Response)

5. Is the manuscript presented in an intelligible fashion and written in standard English?

Reviewer #1: Yes

Reviewer #2: (No Response)

6. Review Comments to the Author

Reviewer #1: Dear authors and editors,

As far as I can tell all my comments and concerns have been addressed.

All the supplementary material is now sufficiently documented to reproduce the results.

Figures and the analysis description now match.

The wording of the discussion has been appropriately hedged.

I have no further request to make.

I thank the authors for their patience.

Best regards

Reviewer #2: (No Response)

7. PLOS authors have the option to publish the peer review history of their article (what does this mean?). If published, this will include your full peer review and any attached files.

Reviewer #1: **Yes: **Remi Gau

Reviewer #2: No

---

## [Author Response · Author response to Decision Letter 2]

11 Aug 2020

Dear Dr. Dr. Naudet,

we would like to thank you for your additional comments on our revised manuscript entitled „A Preregistered Multi-Lab Replication of Maier et al. (2014, Exp. 4) Testing Retroactive Avoidance” submitted for publication to PLOS ONE. We tried to carefully respond to the comments you made concerning the abstract. In this document you will find our answers to your comments and a description of how we adjusted the text (see **). 

Thank you and the reviewers again for the open and fair scientific discussion we had and your profound work on this paper.

Comment of the editor

I propose to use a structured abstract (i.e. more focused on the study methods and results).

** We shortened the theoretical part at the beginning and focused more exclusively on „retroactive avoidance“. We also described the design and the results in more detail.

Importantly, I ask you to delete any discussion about post-hoc analyses in the abstract.

**We deleted any discussion about the post-hoc findings.

I propose to add a few words in the abstract to highlight the main strengths and also limitations of the paper.

**We highlighted the strengths and limitations of the study at the end of the abstract.

Sincerely,

Markus Maier et al.

---

## [Editor Report · Decision Letter 3]

17 Aug 2020

A Preregistered Multi-Lab Replication of Maier et al. (2014, Exp. 4) Testing Retroactive Avoidance

PONE-D-20-02005R3

Dear Dr. Maier,

We’re pleased to inform you that your manuscript has been judged scientifically suitable for publication and will be formally accepted for publication once it meets all outstanding technical requirements.

Kind regards,

Florian Naudet, M.D., M.P.H., Ph.D.

Academic Editor

PLOS ONE
---

## [Editor Report · Acceptance letter]

20 Aug 2020

PONE-D-20-02005R3 

A Preregistered Multi-Lab Replication of Maier et al. (2014, Exp. 4) Testing Retroactive Avoidance 

Dear Dr. Maier:

I'm pleased to inform you that your manuscript has been deemed suitable for publication in PLOS ONE. Congratulations! Your manuscript is now with our production department. 

Kind regards, 

on behalf of

Pr. Florian Naudet 

Academic Editor

PLOS ONE